

# 1 Assimilating High-resolution Sea Surface Temperature Data

# 2 Improves the Ocean Forecast in the Baltic Sea

Ye Liu[1], Weiwei Fu[2]
1. Swedish Meteorological and Hydrological Institute, Norrköping 60176, Sweden.
2. Department of Earth System Science, University of California, Irvine, California, 92697, USA.
*Correspondence to*: Ye Liu (ye.liu@smhi.se)
**Abstract.** We assess the impact of assimilating the satellite sea surface temperature (SST) data on the Baltic
forecast, practically on the forecast of ocean variables related to SST. For this purpose, a multivariable data
assimilation (DA) system has been developed based on a Nordic version of the Nucleus for European Model-
ling of the Ocean (NEMO-Nordic). We use a localized Singular Evolutive Interpolated Kalman (SEIK) filter to
characterize correlation scales in the coastal regions. High resolution SST from OSISAF is assimilated to verify
the performance of DA system. The assimilation run shows very stable improvements on the model simulation
as compared with both independent and dependent observations. The SST prediction of NEMO-Nordic is sig-
nificantly enhanced by the DA system. Temperatures are also closer to observation in the DA system than the
model results in the water above 100 m in the Baltic Sea. In the deeper layers, salinity is also slightly im-
proved. Besides, we find that Sea level anomaly (SLA) is improved with the SST assimilation. Comparison
with independent tide gauge data show that overall root mean square error (RMSE) is reduced by 1.8% and
overall correlation coefficient is increased by 0.4%. Moreover, the sea ice concentration forecast is improved
considerably in the Baltic proper, the Gulf of Finland and the Bothnian Sea, respectively.



# 1. Introduction

Monitoring the marine status of the Baltic Sea with relevant resolution and accuracy is a key requirement
to serve the marine policy for detecting the influence of human activities on the environment and better under-
standing the response of ocean to accelerating global climate change. The Baltic Sea is one of the largest brack-
ish seas in the world. It is a semi-enclosed basin, whose hydrography is highly variable and influenced by
large-scale atmospheric processes and significant influx of freshwater from rivers runoff and precipitation
(Leppäranta and Myrberg, 2009). In addition, the water exchange between the North Sea and Baltic Sea
through the Danish straits is hindered by shallow topographic restrictions in the transition zone (Fig. 1).
A characteristic feature of numerical forecast in the Baltic Sea is in itself a major challenge because of
complex topography and rich dynamics. A number of operational ocean forecasting systems for the Baltic Sea
have been developed using hydrological model by operational agencies around this region. Traditionally, these
operational models have a horizontal resolution of 1–5 km and approximately 20–100 layers in vertical struc-
ture. Due to the geographic location and conditions of the Baltic Sea, even higher resolutions are often needed
to better understand the circulation dynamics. However, even ocean circulation models with a particularly high
spatial resolution (e.g. 1 km) cannot resolve all dynamically important physical processes in the ocean
(Malanotte-Rizzoli and Tziperman, 1996). In general, the forecast quality for a numeric model depends on ini-
tial conditions, boundary conditions (lateral, open boundaries as well as meteorological forcing and bathyme-
try) and a robust numerical model itself. As an operational forecasting agency, the Swedish Meteorological and
Hydrological Institute's (SMHI) needs to issue well-informed forecasts and warnings for decision making by
other authorities during e.g. severe weather events, but also to the public. To improve the forecast quality, the
core three-dimensional dynamic model of the SMHI operational forecast system has recently migrated to the





Nordic version of the Nucleus for European Modelling of the Ocean (NEMO-Nordic).

In additional to model development, an extended observational network has been established by the

joints efforts of the countries surrounding the Baltic Sea. The observation platforms include vessels, buoys,
coastal stations, satellite, etc. Specially, the observations from satellite have dominated the coverage of SST
observational networks in the Baltic Sea (She et al. 2007). Among satellite products, the SST is most popularly
and widely used to the operational forecast, reanalysis or validation of the model because of both its coverage
and properties. SST acts as a medium between atmospheric and oceanic variations through activation of cou-
pling mechanisms. SST is also a key ocean variable to link many processes that occur in the upper ocean, for
example, air-sea exchange of energy, primary productivity, and formation of water masses (Tranchant et al.,

2008).

A realistic forecast of SST is essential to an ocean forecasting system. SST is especially important for the

Baltic Sea that the average water depth is only 56 m and its surface water is directly related to the bottom water
by the mixing in the shallow sub-basins. Recently, the applications of SST for forecasting and analyzing the
status of the North Sea and Baltic Sea have received particular attention. In the short-term forecast, Losa et al.
(2012, 2014) investigated the systematic model uncertainties for forecasting the North and Baltic Seas by as-
similating the Advanced Very High Resolution Radiometer (AVHRR) SST data. Nowicki et al. (2015) applied
SST observed from Aqua Moderate Resolution Imaging Spectroradiometer (MODIS) into 3D coupled ecosys-
tem model of the Baltic Sea with the Cressman analysis scheme. O'Dea et al. (2016) enhanced the SST predic-
tion skill of the operational system by assimilating both in-situ data and level 2 SST data provided by the Glob-
al Ocean Data Assimilation Experiment High-Resolution SST (GHRSST) into a European North-West shelf
operational model. Moreover, SST has been used in the long-term analysis in this region. For instance,
Stramska and Bialogrodzka (2015) analyzed spatial and temporal variability of SST in the Baltic sea based on
32-years of satellite data, which indicate that there is a statistically significant trend of increasing SST in the



entire Baltic sea. However, these long-term SST data haven't been used to verify the application of sophisticat-
ed DA methods for hydrography model in the Baltic profiles simulation, especially at the Baltic deep water
regions. Another important question is: what amount of satellite SST can improve long-term forecast of ocean
variables related to SST in the Baltic Sea.

The objective of this study is to address the impact of assimilating a high resolution SST product on the

forecast of the Baltic Sea, particularly the forecast of SST related variables like sea level and sea ice. It is also
the first time that satellite SST from OSISAF was assimilated into NEMO-Nordic model (NEMO variant for
the North Sea and Baltic Sea). For operational forecast, the SST from the Ocean and Sea Ice Satellite Applica-
tion Facility (OSISAF) is the most important dataset in the Baltic Sea. Therefore, exploring the potential of this
product is critically important to further improving the new operational forecast system. In addition, our study
will enrich the reanalysis database of the Baltic Sea. In this study, we use the Singular Evolutive Interpolated
Kalman (SEIK) filter (Pham, 2001) to account for the model uncertainties arising from a wide range of spatial
and temporal scales (Haines, 2010). One of our focuses is the impact of SST on the modeled sea level and the
sea ice in the Baltic Sea. For the whole Baltic Sea, how the SST assimilation influences the temperature and
salinity (T/S) on the different depth is another focus of this study.

The outline of the paper is as follows: the model configuration and SEIK scheme are described in Sec-

tion 2. An overview of the observations used in this study is presented in Section 3. The implementation of DA
experiment is given in section 4 together with the sampling of ensemble and localization. Results are compared
with observations for temperature, salinity, sea level Anomaly and sea ice in Section 5. In this section, the im-
pact of data assimilation on the forecasts is also investigated. Conclusions and discussions are given in section

6.


**2. Methodology**





## 2.1 NEMO-Nordic

NEMO (Nucleus for European Modelling of the Ocean; Madec, 2008) has been set up at SMHI for the North Sea and the Baltic Sea, a configuration called NEMO-Nordic (Hordoir et al., 2015) (Fig. 1). Open boundaries are implemented in northern North Sea between Scotland and Norway and in the English Channel between Brittany and Cornwall, respectively (Hordoir et al., 2013). In this study, NEMO-Nordic employs a horizontal resolution of 2 nautical miles (3.7 km) and 56 vertical levels, and with a vertical resolution of 3 m close to the surface, decreasing to 22 m at the bottom of the deepest part of the Norwegian trench. NEMO-Nordic uses a fully nonlinear explicit free surface (Adcroft and Campin, 2004). A bulk formulation is used for the surface boundary condition (Large and Yeager, 2004). The ocean model is coupled to the Louvain-la-Neuve Sea Ice Model (LIM3) sea ice model (Vancoppenolle et al., 2008) with a constant value of $10^{-3}$ PSU for the sea-ice salinity. A time-splitting approach is used to compute a barotropic and a baroclinic mode, as well as the interaction between them. A Tidal Inversion Model is used to define the barotropic mode at the open boundary conditions (Egbert and Erofeeva, 2002). 11 tidal harmonics are defined for sea level and barotropic tidal velocities. In addition, a coarse resolution barotropic storm surge model covering a large area of the Northern Atlantic basin provides wind-driven sea level that is added to the tidal contribution. The TS data at the open boundary are provided by the Levitus climatology (Levitus and Boyer, 1994). Radiation conditions are applied to calculate baroclinic velocities at these boundaries. A quadratic friction is applied with a constant bottom roughness of 3 cm, and the drag coefficient is computed for each bottom grid cell. NEMO-Nordic uses a TVD advection scheme with a modified leapfrog approach that ensures a very high degree of tracer conserva-tion (Leclair and Madec, 2009). Unresolved vertical turbulence is parameterized with $\varkappa$-$\varepsilon$ scheme (Umlauf and Burchard, 2003). In addition, Galperin parameterization is used to obtain a stable long-term stratification for the Baltic Sea (Galperin et al., 1988).

A Laplacian isopycnal diffusion is used for both momentum and tracers with a diffusion parameter that is





constant in time, but varies in space. Additional strong isopycnal diffusion is used close to the Neva river in-
flow (Gulf of St. Petersburg) in order to avoid negative salinities. The bottom boundary layer is parameterized
to ease the propagation of saltwater inflows between the Danish Straits and the deepest layers of the Baltic Sea
(Beckmann and Doscher, 1997). A free-slip option is used for lateral boundaries.

The model is forced by meteorological forcing derived from a downscaled run of Euro4M reanalysis

(Dahlgren et al., 2014). The downscaling is based on the regional atmospheric model RCA4 (Samuelsson et al.,
2011) which uses the reanalysis data as boundary conditions. A runoff database provides the river flow to
NEMO-Nordic; it includes inter-annual variability for the Baltic Sea basin and is based on climatological val-
ues for the North Sea basin. The salinity of the river runoff is set to a constant value of $10^{-3}$ PSU, which is the
same value used for the sea-ice to avoid any negative salinity.

**2.2 Sigular Evolutive Interpolated Kalman (SEIK) filter**

The method used to assimilate SST into NEMO-Nordic is the Singular Evolutive Interpolated Kalman

(SEIK) filter (Pham et al., 2001). This is a sequential data assimilation scheme, which is an error subspace ex-
tend Kalman filter that uses a minimum number of ensemble members to reduce the prohibitive computation
burden (Pham, 2001). The SEIK filter proceeds in correction and forecast step:
1. Forecast: the analysis state $\boldsymbol{X}^a$ at time $t_{i-1}$ is integrated forward to the time of the next availabe observations
$t_i$ to compute the forecast state $\boldsymbol{X}^f$,

$$\boldsymbol{X}^f(t_i) = \boldsymbol{M}(t_{i-1}, t_i)\boldsymbol{X}^a(t_{i-1}) \qquad (1),$$

where $\boldsymbol{M}$ denotes the nonlinear dynamic model operator that integrates a model state from time $t_{i-1}$ to time
$t_i$. The superscript '$f$' and '$a$' denote the forecast and analysis. The corresponding error covariance matrix can
be expressed as:

$$\boldsymbol{P}^f(t_i) = \boldsymbol{L}_i[(r+1)\,\boldsymbol{T}^T\boldsymbol{T}]^{-1}\boldsymbol{L}_i^T + \boldsymbol{Q}_i \qquad (2),$$

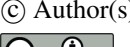



$$L_i = X^f(t_i)T \tag{3},$$

with $Q_i$ being the covariance matrix of model uncertainties and $r + 1$ is the minimum number of sample en-
semble members for error covariance matrix. The superscript $'T'$ denotes the transpose of matrix. The full rank
matrix $T$ has a dimension of $(r + 1) \times r$ with zero column sums.
2. Correction: when the observation is available at time $t_i$, the SEIK filter merged the information from model
and observation to produce the analysis state with the formula:
$$X^a(t_i) = X^f(t_i) + K_i[Y^o(t_i) - H_i X^f(t_i)] \tag{4}.$$

Here $Y^o$ is a vector of observations. The gain matrix $K$, which linearly interpolates between the observations
and the forecast, is given by
$$K_i = P_i^f H_i^T \left( H_i P_i^f H_i^T + R_i \right)^{-1} = L_i U_i (H_i L_i)^T R_i^{-1} \tag{5},$$

where $H_i$ denotes the linearization of observation operator, which mapping the model space to the observation
space. $R$ is the observation error covariance matrix. The matrix $U_i$ is updated according to
$$U_i^{-1} = [U_{i-1} + (L_i^T L_i)^{-1} L_i^T Q_i L_i (L_i^T L_i)^{-1}]^{-1} + L_i^T H_i^T R_i^{-1} H_i L_i \tag{6}.$$

A second-order exact sampling is used to initialize the SEIK filter. At time $t_{i-1}$, a analysis state $X^a(t_{i-1})$ and
its corresponding error covariance matrix $P^a(t_{i-1})$, in the factorized form $L_{i-1} U_{i-1} L_{i-1}^T$ , are available. The
samples can be given by the following formular:
$$X_k^a(t_{i-1}) = \overline{X^a}(t_{i-1}) + \sqrt{r+1} L_{i-1} (\Omega_{k,i-1} C_{i-1})^T \tag{7}.$$

For $1 \leq k \leq r + 1$, the $C_{i-1}$ is the Cholesky decomposition of $U_{i-1}^{-1}$ and $\Omega_{i-1}$ is a $(r + 1) \times r$ matrix with or-
thonormal columns and zero column sums, where $\Omega_{k,i-1}$ denotes the $k^{th}$ row of $\Omega_{i-1}$. $\overline{X^a}$ is the average of the
analysis state.

**3. Observations**



### 3.1 Satellite observations

The satellite SST used in DA was provided by OSISAF (http://osisaf.met.no/p/sst/index.html). OSISAF products are using in priority the European Meteorological satellites METEOSAT and MetOp and also several American satellites operated by NOAA, DMSP and NASA. Its aim is to produce, control and distribute operationally in near real-time products using available satellite data. The satellite datasets product used here includes the observations from polar orbiting satellites (the EUMETSAT MetOp-A and NOAA-18, -19) with the AVHRR instrument. The SST product has a resolution of 5 km and is produced twice daily at 00 UTC and 12 UTC. It covers the Atlantic Ocean from 50°N to 90°N. The SST observations are thermal infrared observations from the AVHRR instrument and are therefore limited by cloud cover (Kilpatrick et al. 2001). The cloud mask in use is based on a multi-spectral thresholding algorithm by SMHI. The products were retrieved using a non-linear split window algorithm (Walton et al. 1998). The coefficients in the retrieval algorithm are determined through regression toward in situ observations, and the dataset thus represents the subskin temperature of the oceans. Further, subskin observations are subject to diurnal warming effects, which can be significant in the Baltic Sea. Here only the subskin SST at night, which is comparable to in situ (buoy) measurement, is used to minimum this effect. The SST is controlled with the climatology check. A quality level from 0 to 5 is associated with every pixel. The higher the level value, the better the quality of the observations (Brisson et al., 2001). Observations with quality level 4 (good) or 5 (excellent) are collected for the analysis and observations of low quality were removed. By applying the above quality control processes, only a subset of the original OSISAF products is kept in this study. Based on the former validation, a bias value of 0.5°C is given for this product.

Further, the IceMap from a sea ice concentration dataset with a high spatial resolution of 5 km (http://www.smhi.se/oceanografi/iceservice/is_prod_en.php) is used to validate the DA results. It is produced by SMHI and originates from digitized ice charts. An advantage of this data is that the ice charts are quality checked manually. However, the drawback is that they include some subjective steps. The temporal resolution

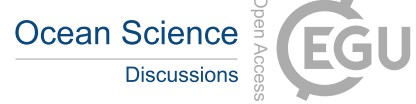


of the IceMap SST is twice a week in the experiment period. Sea ice occurs most frequently in the Bay of
Bothnia, with up to 100 ice covered days per year. However, sea ice can occur in all parts of the Baltic Sea and
Danish straits, demonstrating the need for careful treatment of sea ice in the SST analysis.

**3.2 In situ data**
The observations from the German Maritime and Hydrographic Agency (BSH) moored buoy stations
were collected as independent dataset to validate the assimilation results. The observations have high temporal
resolution and long continuous record. The second dataset was downloaded from the Swedish Oceanographic
Data Centre -SHARK database (http://sharkweb.smhi.se). SHARK mainly contains low-resolution CTD data
from a list of predefined standard stations in the Baltic Sea, as well as in Kattegat and Skagerrak. Only obser-
vations that have passed gross quality control procedures are collected into the SHARK database. This proce-
dure includes, for example, location checks and local stability checks. In addition, validating data records from
tide gauges are also used. The sea level anomaly measurements from tide gauges (sea level stations) are meas-
ured in a local height system and values are presented relative to theoretical mean sea level, a level calculated
from many years of annual means, which takes into account the effect of land uplift and sea level rise. The val-
ues are averaged over one hour period.
Not all the available observations from satellite, moored buoys, CTDs, tide gauges were included in this
study. To obtain the high assimilation quality results, another quality control was applied for these data before
they were used into assimilation and validation. These controls include examination of forecast observation
differences by excluding those observations for which the difference between the forecast and the measurement
exceeded given standard maximum deviations. The criteria were set up empirically based on past validation
results of the model (Liu et al. 2013). Furthermore, stations located on land, according to the NEMO-Nordic
grid, were excluded. We also removed the duplicate records of these data.

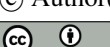



The accuracy of observation error is difficult to be defined for all water points. The observation error
mainly comes from the observation instrument itself, the observation representativeness, the temporary reading
error and imperfect retrieval algorithm. The observation is commonly assumed to be spatially irrelevant, which
results in an error covariance matrix that is time-invariant diagonal and its diagonal elements equal the variance
of observation error. In this study, the observation error was estimated to one value as the sum of all observa-
tion uncertainties used in the analysis. Besides, the uncertainties of satellite SST varies from coast to the open
sea, i.e. higher uncertainties in the coast region relative to the open sea.  We used a constant standard deviation
value of 0.4℃ based on the standard deviation of satellite SST, which ranged from the ~0.1℃ to ~0.5 ℃ in the
Baltic Sea (She et al. 2007, Høyer et al. 2016).

## 4. Configuration of SEIK in the experiment

As above mentioned, the initialization of the filter requires an initial analyzed state and a low rank ap-
proximation of the corresponding estimation of error covariance matrix. The data assimilation process was ini-
tialized by a free model simulation. First the model was spinning up 20 years to reach a statistically steady
state. Then a further (free-run) integration covered the period 2006-2009 was carried out to generate a histori-
cal sequence of model state. To reduce the calculation cost, we took a snapshot in every 6 days and saved 183
state vectors in total to describe the model variability because successive states are quite similar. The initial
ensemble provided an estimate of the initial model state and its uncertainty before the assimilation of SST ob-
servations. The quantity of the model variability was expected to be reasonably comparable with the forecast
error, which was dominated by misplacement of mesoscale features and varies in location and intensity season-
ally. Further, the very high frequencies of model variability were also unfavourable in an ensemble of state
vectors for SST data assimilation (Oke et al., 2005). Therefore, a band-pass filter was used to remove the un-
wanted frequency of model variability. A multivariable Empirical Orthogonal Functions (EOF) analysis was

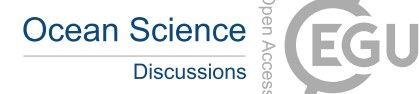

applied to compute the initial low rank error covariance matrix. There does not exist uniform nature of error
covariance for the variables of the model state vector and for the coastal zones in the North Sea and Baltic Sea.
Each state variable was then normalized by the inverse of its spatially averaged variance at every model layer.
At last, 34 leading EOF modes were kept and they explained 85% overall variability. Then the initial error co-
variance matrix was estimated by $P^a(t_0) \approx L_0 U_0 L_0^T$ , where the $L_0$ is composited by the leading EOF modes
and $U_0$ is diagonal matrix with the corresponding eigenvalues on its diagonal.

A forgotten factor $\rho$ was introduced to parameterize the imperfect model by amplifying the already existing

modes of the background error. The matrix $U_i$ was calculated by
$$\boldsymbol{U}_i^{-1} = \rho(r+1)\boldsymbol{T}^T\boldsymbol{T} + \boldsymbol{L}_i^T\mathbf{H}_i^T\boldsymbol{R}_i^{-1}\mathbf{H}_i\boldsymbol{L}_i \qquad (8)$$
Further, localization was used to remove the unrealistic long-range correlation with a quasi-Gaussian function
and a uniform horizontal correlation scale (Liu et al. 2013). It was performed by neglecting observations that
were beyond correlation distance from an analyzed grid point and only data points located in the "neighbor-
hood" of an analyzed grid point should contribute to the analysis at this point. As a result, the quality of fields
obtained by data assimilation was determined by the observations coverage and quality (Liu et al. 2009).

The localization scale is another import factor to the assimilation system, especially at the coastal region.

Large correlation scale may transfer artificial increments to the positions far away from the analysis observa-
tion during the DA process. However, small correlation scale is prone to cause the singularity of ocean state
around analyzed observation and break the continuity of the ocean state. Hence, an unreasonable scale causes
the instability of the model integration or degrades the assimilation quality. Unfortunately, the accuracy length
for the correlation is unknown for the North Sea and Baltic Sea. The correlation length scale, to some extent,
depends on the dynamics of the basin, more precisely, on the Rossby radius of deformation, which varies from
~ 200 km in the barotropic mode to ~ 10 km or even less in the baroclinic mode. According to the former re-





searches like Liu et al. (2013, 2017), a length scale of 70 km was specified for both the North Sea and Baltic
Sea in this study. Not that this value may be not perfect and more accurate correlation length needs to be tested
for SEIK. For example, spatially variable length scales are the next step for the regional DA simulations.

**5. Results**

In the following sub-sections, we conducted two runs with and without assimilation of the SST obser-

vations from the OSISAF database, both runs with the above setup of the analysis system. Accordingly, the
runs with and without assimilation are called ASSIM and FREE, respectively. We considered the evolution of
SST based on 48-hourly local analysis from 1 January 2010 to 31 December 2010. The SST simulation in two
runs was assessed with dependent and independent observations. Then we analyzed the impact of the data as-
similation on the profile simulation of T/S. At last, we evaluated the system performance with respect to sea
surface height and sea ice, respectively.

**5.1 Comparison with satellite data**

First, we presented a verification showing how the assimilation of the OSISAF SST data works, and

what impact it has on the model SST in the Fig. 2. The first case was given at 11 January 2010, a date with
clear weather and many observations available, which represents a winter situation. The model has obvious
difficulties in reproducing the observed SST. The cold biases in the forecast were found in the Skagerrak, west
coast of the Baltic proper and the Bothnian Bay, respectively. However, the warm biases appeared in the inte-
rior of the Baltic Sea and the Kattegat. The largest deviation in the FREE reached 2.2 ℃ at the Skagerrak. Ap-
parently, temperature by assimilation analysis agreed with the satellite-derived data much better. This correc-
tion at the analysis step has allowed us to reduce the deviation of the SST forecast from the observations. The
SST bias of model forecast possibility has seasonal variability because of the errors in the forcing and/or heat

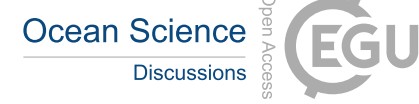

flux parameterization used in the ocean model (Fu et al. 2012). Therefore, the DA system simulation was also
verified in summer on 2 June 2010, which has also many available OSISAF observations. The biases on 2 June
2010 were obviously different from that on 11 January 2010. Moreover, it was found they had a roughly oppo-
site bias signal. For example, relative to the OSISAF SST at the Baltic proper, Bothnian Sea and Bothnian Bay,
FREE produced relatively warmer water in winter and colder water in summer (Fig. 2), respectively. After data
assimilation, the analysis increments were appropriately added to the model field. In general, the SST DA has
improved the model SST forecasts in both winter and summer (Fig. 2).

Maps of annual averaged RMSE of SST from two runs relative to the IceMap observation are shown in Fig.

3.  Obviously, the RMSE in FREE and ASSIM had different distribution in the Baltic Sea. In general, FREE
had smaller error in the Skagerrak, eastern the Kattegat and the interior of the Bothnian Sea relative to other
subbasin of the Baltic Sea.  The largest RMSE was found at the connection region between the Baltic proper
and the Bothnian Sea. This could be caused by the shallow water, complicated bathymetry and large observa-
tion biases in this area. It was also noted that the RMSE was larger in the coast region compared to its interior
in the Baltic proper and Bothnian Sea. After the assimilation, the SST has been significantly improved. The
RMSE of SST from ASSIM was generally smaller than 1.0 ℃. However, there were still some regions where
the improvements were relatively small and the RMSE of SST was greater than 1.0 ℃. These large errors were
predominantly located at the edge of the Baltic Sea and the Danish straits. For instance, the RMSE of SST was
greater than 1.2 ℃ at both the entrance of the Gulf of Finland and the west coast of the Bothnian Sea. The rela-
tively small improvements were regularly caused by the rare observations or the less accurate observations near
the coast water.

The overall daily averaged SST errors against the IceMap observations have been estimated (Fig. 4). The

observations had better coverage in summer and autumn than in winter and spring. The variability of the num-
ber of observation directly affected the assessment of DA results. The model biases had pronounced seasonal





variability, which had small values in spring and winter. In general, the assimilation provided better SST esti-
mations. The free run had a RMSE of 1.47 ℃. After the assimilation, the RMSE was reduced to 1.03 ℃,
whereas the bias was reduced by 0.73 ℃. An interesting feature was that the SST error reduction due to the
assimilation was almost consistent with the variability of the number of IceMap observations. For example, the
improvement became large with increasing the number of IceMap observations from March to June 2010.
However, the number of observations was kept constant during the period June-November 2010 and the im-
provement shown in both the bias and RMSE of SST did not exhibit large variability, which meant reliable per-
formance of the DA system.

**5.2 Comparison with independent in-situ data**
The time series of T/S were compared with independent observations located at Arkona station
(13.87ºE, 54.88ºN) in the Arkona Basin and at BY15 (20.05 ºE, 57.33 ºN) in the Eastern Gotland Basin, respec-
tively. These two stations were selected to verify the experiment results because of their relatively completed
observation records for the experiment period. In the Arokna Basin, the water depth was shallow and the water
column can be well mixed between surface and bottom water. Thus, the bottom T/S was largely affected by the
surface dynamic (Liu et al. 2014). Relative to observations, the model had warm biases at this station (Fig. 5).
The temperatures differ by about 15–22 ºC between summer and winter. At a depth of 25m, the observed tem-
perature showed the largest variability, which was a good representation of the bottom characteristics of the
mixed layer. In mid-August, the temperature was abruptly increased by 10ºC at a depth of 25m and slightly
decreased at surface, respectively. The reason perhaps the surface water suddenly sinks to deeper layers, which
warm the deep water. However, this dynamic process hasn't reached to Arkona bottom and it didn't cause the
obvious bottom temperature variability. Both FREE and ASSIM had reproduced this process, whereas FREE
showed larger temperature biases. To the salinity at the Arkona station, the surface observations were missing,





the comparison at 7 m depth verified the subsurface simulations. The observations showed larger salinity vari-
ability in winter relative to summer. This pronounced seasonal variation is associated with the variation of
fresh river runoff and net E–P (Evaporation–Precipitation) flux (Fu et al, 2012). At a depth of 7 m, salinity was
obviously underestimated from April to September and overestimated after November although the ASSIM
had slightly better results compared to FREE. The DA also provided better simulation of salinity at 25 m depth.
For example, the salinity bias in the October was reduced by 3 psu by DA. At a depth of 40 m, the saltwater
inflows were observed, resulting in sudden increases of salinity. For instance, the salinity was increased by 3.5
psu in February followed by a decreasing trend. The variations were reproduced in both FREE and ASSIM,
whereas the intensity of the decreased process is weakly simulated with a difference of 3 psu and the inflow in
March was not strong enough relative to the observed one. Observations also showed a large salinity variability
amounts to 4–8 psu in the autumn. Although FREE and ASSIM had shown these changes, their magnitude was
obvious weaker than observations. The possibility reason was that the model's resolution was inadequate to
well resolve the topography and eddies in this area. Both the large runoff and the complicated bathymetry
posed challenges for the model to tackle the small-scale dynamic process in such a shallow basin. A higher
resolution model perhaps was more preferable to study this dynamic process.

The Eastern Gotland Basin has deeper water depth compared to the Arokan Basin, in which the water

column is permanently stratified and the halocline lies at about 60–80 m (Fu et al, 2012). The mixing and sink-
ing of T/S are hindered by the strong stratification. Unlike observations in the Arokna Basin (Fig. 5), the CTD
observations at BY15 had lower temporal resolution with almost one observation per month. In the mixing lay-
er, it can be seen model had overestimated the temperature (Fig. 6). At a depth of 10 m, ASSIM has remarkably
improved the simulation of temperature relative to FREE. The bias has been reduced by 3℃ in the spring of
2010. At 175 m depth, observed temperature showed very small variation. The reason was that the main source
for deep water ventilation is the saltwater inflows which are suppressed by runoff within a depth range of 75–





135 m in the Eastern Gotland Basin (Vali et al. 2013). As a result, updating the bottom water is very slow. Both
FREE and ASSIM overestimated the temperature in the spring and the beginning of summer of 2010. Further,
ASSIM has increased the temperature bias after mid-summer relative to FREE.  This result might be explained
by that the strong correlation isn't expected between surface and layers bellow the halocline because of the
strong stratification in this basin, which perhaps yield the artificial correction. Therefore, the improvement of
the surface temperature cannot guarantee its positive influence on the bottom temperature. To the salinity, the
model had less accurate simulation with generally low salinity biases at 10 m depth. ASSIM provided better
salinity simulation compared to FREE. At 70 m depth, the small variation of salinity was found after DA.
Moreover, at 175 m depth, the observation had very small variability about 0.1 psu. In general, both experi-
ments have reproduced these variations. However, FREE increased salinity by 0.2 psu from March to April
relative to the observation, which caused the overall salinity overestimated amount to 0.2 psu. This increasing
process wasn't shown in observations and the reason remained unclear. The DA has shown slight improve-
ment, but it still salter than the observations.

Further, all the SHARK observations have been used to calculate the RMSE of T/S. The temporal and spa-

tial distribution of these observations is shown in Fig. 7. These observations were unevenly distributed in the
Baltic Sea. In the Skagerrak, the observations appeared at the Danish and Swedish coast. However, in the
Bornholmn Basin, Kattegat, and Baltic proper, the observations mainly were found in the central and the Swe-
dish coast side. There were also many observations in the Bothnian Sea and rare observations in the central of
the Bothnian Bay. It must be noticed that there aren't SHARK observations in both the Gulf of Finland and
Gulf of Riga during the experiment period. Moreover, these SHARK profiles in the first four months were
mainly located from the Skagerrak to the Baltic proper, which are relatively rare in the northern Baltic Sea. In
the Bothnian Bay, the observations are mainly in the winter period.  Figure 8 shows the RMSE of T/S as a
function of water depth. The overall RMSE of T/S was decreased from 1.37 and 0.92 psu for FREE to 1.21 ℃

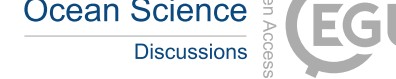



and 0.94 psu for ASSI, which means the T/S have been improved 11.68% and -2.17%, respectively. Maximum
and minimum RMSEs of temperature were 1.86 and 0.79 ℃ for FREE and 1.76 and 0.84 ℃ for ASSIM, whilst
these values corresponded to the upper layer and close to the bottom, respectively. With increasing depth, the
RMSE of T/S was decreasing below 70 m depth. The temperature in ASSIM was significantly improved from
the surface to the halocline. The improvement of temperature was decreased with depth. The experiment
showed that ASSIM slightly degraded the temperature in deeper layers and the salinity above the halocline,
respectively.

**5.3 Sea Level Anomaly**

SLA represents a vertically integrated effect of the T/S variations over the whole water column. The

accurate simulation of SLA is thus a good indicator of the model performance. Therefore, validating the impact
of SST assimilation on the simulation of SLA is very important to the Baltic Sea forecast. The observations
from the 24 tide gauge stations were used. These gauge stations are mainly located at the Swedish coast. The
correlation coefficients and RMSE were calculated between the observation and two simulations (Table 1).
Since only the SST is assimilated in this study, the SLA observations are completely independent.

Compared with tide gauge observations, the SLAs in FREE have been well simulated at these stations. The

correlation coefficients in FREE at 21 stations are all larger than 0.9. At the Klagshamn, Barseback and Viken
stations, the coefficients are smaller than 0.9 but still greater than 0.86. These three stations are located near the
Sound where the sub-grid scale feature of narrow transport cannot be fully resolved even in a high resolution
model. Besides, the RMSE of SLA is smaller than 0.1 m at 19 stations. Generally, SST DA had positive effects
on the simulation of SLA. In Table 1, the SLAs in ASSIM were better correlated with the tide gauge data than
that in FREE. All the RMSEs were reduced in ASSIM relative to FREE. The overall RMSE and correlation
coefficients were reduced by 1.8% and increased by 0.4%, respectively. These stations are located from north



to south of the Swedish coast, which imply the ASSIM had reliable performance in the all Swedish coast (Ta-
ble 1). In addition, the time series of simulated SLA bias (simulation minus observation) at four stations were
selected to evaluate the simulation results (Fig. 9). These four stations were selected to represent the model per-
formance at different positions of the Swedish coast. Model showed evidently different performance in these
four stations. Briefly, model produced the larger SLA than observed ones. The largest biases appeared at the
Simrishamn station where model had smaller SLA relative to the observation. Moreover, the variation of SLA
bias at the Smogen station had higher frequency compared to other stations. The reason was that the Smogen
station was located at the transition zone where the water had higher frequency variations caused by the brack-
ish Baltic in/outflowing relative to other three stations.  It was also found that modeling biases at LandsortNor-
ra had smaller variability compared to other stations. ASSIM reduced these biases by 3–8% at these four sta-
tions. These improvements were mainly in later spring and summer. The SST assimilation had less impact in
late winter and early spring compared to other seasons. Besides, the impact of SST assimilation on SLA simu-
lation was not same in the four positions. For instance, the SLA discrepancies between ASSIM and FREE was
smaller at LandsortNorra relative to that at Smogen and ASSIM slightly degraded the SLA simulation relative
to FREE at both LandsortNorra and Ratan after October. This phenomenon was possibly caused by the imper-
fect correlation between SST and SLA in the stationary samples. Further, these steric small changes of SLA by
data assimilation in Table 1 were what we expected because only SST was assimilated into Nemo-Nordic.

**5.4 Sea ice**

Sea ice in the Baltic Sea occurs primarily in its north region and influences the Baltic climate. Accurate

detecting the sea ice is very useful to the northern Baltic living because too much or too little sea ice can be a
problem for wildlife and people. Sea ice concentration (SIC) is an important and common indicator to model-
ing sea ice environment. We assessed the SIC from simulations against the IceMap observations in Fig. 10.



Differ from the daily evaluation in Losa et al. (2014), the monthly mean SIC was used to represent the general
status of sea ice in the Baltic Sea. Besides, SIC in January, February and December showed the variation of the
sea ice in winter.

In January 2010, the observations showed large ice coverage in the Bothnian Bay and the Gulf of Fin-

land and small SIC in the Gulf of Riga, respectively. Model generally reproduced this distribution of sea ice.
However, FREE simulated too much sea ice in the Gulf of Finland and the eastern coast of the Baltic proper
relative to observations. For example, SIC from FREE almost to 30% higher than observations along the Esto-
nia coastline. It could be seen that the SST DA reduced these biases. The reason is the SST DA modified the
thermal expansion by providing the well temperature fields above the thermocline. The temperature in Febru-
ary became colder relative to January in the Baltic Sea. As a result, the sea ice in February extended to the
Bothnian Sea and the whole Gulf of Riga. Observation also showed small SIC in Kattegat and Skagerrak.
Model simulated higher SIC in the Bothnian Sea with largest biases along the Swedish and Finnish coast. As
an example, the observed ice in the Bothnian Sea was characterized by concentrations mainly smaller than 0.5,
whereas modeled ice in FREE had concentration greater than 0.9 in the shallow region of the Bothnian Sea.
FREE also had smaller ice coverage with lower SIC in the transition zone between the North Sea and the Baltic
Sea relative to IceMap. After the SST assimilation, ASSIM reduced SIC in the Bothnian Bay and the west
coast of the Baltic Sea, which was closer to the observations. The ice in ASSIM didn't have obvious variation
in Kattegat and Skagerrak yet. ASSIM also reduced too much ice at the southern of the Bothhomn Basin. The
reason is that the satellite SST observations had limited accuracy near the coast and they could bring artificial
information into the modeling. In December, sea ice coverage was smaller because of relatively warm tempera-
ture compared to that in other winter month. Most of the sea ice with high concentration was observed at the
edge of the Bothnian bay. Nevertheless, high concentration ice in FREE also happened at the transition zone
between the Bothnian Sea and Bothnian bay. Relatively, ASSIM reduced the high concentration biases of sea



433 ice. By contrast, both ASSIM and FREE had lower concentration ice than observation in the eastern coast of

434 the Bothnian Sea. The SIC from ASSIM was relatively lower than that from FREE in the northern Finish coast,

435 whereas the observations had high concentration ice there.


437 ## 6. Conclusion and discussions

438 A DA system based on a localized SEIK filter has been coupled to the NEMO circulation model of the

439 North and Baltic Seas. The method was successfully applied for assimilating high resolution satellite SST data.

440 We demonstrated that, over the period of 2010, the agreement of the SST forecast with the independent satel-

441 lite observation was improved by ∼ 29.93% in comparison with the regular forecast without DA. The assimila-

442 tion quality is directly related to the number of observation.

443 Compared with independent in-situ data from SHARK, the results showed the overall RMSE of T/S

444 was reduced by 11.68% and decreased by 2.17%, respectively. These variations of T/S mainly occurred in the

445 water above 100 m. However, in the deeper layers, the temperature was slightly degraded while salinity was

446 slightly improved. This is partially caused by the artificial correlation between surface layer and deeper layers.

447 The improvement of temperature by SST DA can't guarantee corresponding improvement of the salinity. Both

448 ASSIM and FREE have captured the main dynamic process in the Baltic Sea, for example, the inflow and the

449 sink. However, ASSIM is closer to the observed one relative to FREE.

450 Further validation by the independent SLA observations shows that all RMSEs and correlations for all

451 21 stations are smaller than 0.12 m and greater than 0.86, respectively. After DA, the SLAs at these stations

452 have been slightly improved. In general, the RMSE and correlation coefficients of SLA were reduced by 1.8%

453 and increased by 0.4%, respectively. Further, the model-observation comparison at selected four stations indi-

454 cates that these improvements are mainly in later of spring and summer. The comparisons also denote the SST

455 assimilation has less impact in the late winter and early spring relative to other seasons. When compared with

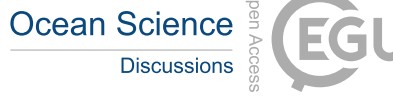

monthly mean observations of SIC, both assimilation run and free run reproduced main distributions of sea ice
in the Baltic Sea. The SST assimilation has significantly improved the results of sea ice from FREE in the Gulf
of Finland, the Bothnian Sea and eastern coast of the Baltic proper. However, minor improvements were found
in Kattegat and Skagerrak.

The results of the SST assimilation are encouraging and the assimilation helps to ameliorate some

model deficiencies such as the simulation of sea ice in the Gulf of Finland. However, some problems need to
be further addressed in the SST DA in the future: firstly, the SST assimilation has worse influence on the simu-
lation of salinity in the upper layers and temperature in the deeper layers. Losa et al. (2012) denoted that the
salinity simulation quality crucially depends on the assumptions about the model and data error statistics. Here
a stationary ensemble sample was used to represent the correlation between T/S and between surface and deep
water. These relationships could be changed with the varying dynamics and forcing conditions. More sophisti-
cated assumption should be used in the DA of Baltic Sea. Secondly, the SHARK observations in this study are
absent at the Gulf of Finland and Gulf of Riga. This denotes the validation results with SHARK observation
didn't include the evaluation of the simulation of T/S in deep water of these two basins. Thirdly, the univariate
localization scale used in this study could be another problem. The spreading of observation information strong
depended on the correlation scale around the observation position. The large localization scale can introduce
the artificial information, which could degrade the assimilation quality.  A flow-dependent background error
covariance with varying correlation scale may be more appropriate for the Baltic Sea with complex bathymetry
and rich dynamics. Fourthly, the remote sensing observations near the coast could have large bias because of
the limit of the instrument itself. More strict quality controlling method needed to be used for the satellite
coastal observations before their assimilation.

**Acknowledgment**

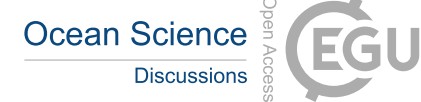




The research presented in this study was funded by the Swedish Space Board within the project 'Assimilating
SLA and SST in an operational ocean forecasting mode for the North Sea and Baltic Sea using satellite obser-
vations and different methodologies' (grant no.172/13).

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





Table 1. The correlation coefficients and RMSE (in m) of the model compared to observed sea level anomaly data in 24 tide gauge stations.

| Station name | Position (degrees) | | FREE | | ASSIM | |
|---|---|---|---|---|---|---|
| | | | Corre. | RMSE | Corre. | RMSE |
| Kalix | 23.09°E | 65.69°N | 0.9603 | 0.1118 | 0.9628 | 0.1109 |
| Furuogrund | 21.23°E | 64.92°N | 0.9577 | 0.1022 | 0.9599 | 0.1014 |
| Skagsudde | 19.01°E | 63.19°N | 0.9641 | 0.0909 | 0.9662 | 0.0901 |
| Ratan | 20.89°E | 63.98°N | 0.9605 | 0.1113 | 0.9641 | 0.1102 |
| Spikarna | 17.53°E | 62.36°N | 0.9673 | 0.0839 | 0.9719 | 0.0822 |
| Forsmark | 18.21°E | 60.41°N | 0.9671 | 0.0639 | 0.9717 | 0.0617 |
| Stockholm | 18.08°E | 59.32°N | 0.9631 | 0.0661 | 0.9672 | 0.0643 |
| Marviken | 16.84°E | 58.55°N | 0.9678 | 0.0668 | 0.9719 | 0.0662 |
| Visby | 18.28°E | 57.64°N | 0.9605 | 0.0746 | 0.9685 | 0.0721 |
| Oskarshamn | 16.48°E | 57.27°N | 0.9595 | 0.0506 | 0.9648 | 0.0477 |
| Kungsholmsfort | 15.59°E | 56.11°N | 0.9554 | 0.0533 | 0.9604 | 0.0507 |
| Simrishamn | 14.36°E | 55.55°N | 0.9084 | 0.0804 | 0.9183 | 0.0765 |
| Skanor | 12.83°E | 55.42°N | 0.9378 | 0.0729 | 0.9409 | 0.0714 |
| Klagshamn | 12.89°E | 55.52°N | 0.8853 | 0.0834 | 0.8856 | 0.0834 |
| Barseback | 12.90°E | 55.76°N | 0.8655 | 0.0795 | 0.8707 | 0.0781 |
| Viken | 12.58°E | 56.14°N | 0.8683 | 0.0989 | 0.8722 | 0.0978 |
| Ringhals | 12.11°E | 57.25°N | 0.9042 | 0.0906 | 0.9087 | 0.0891 |
| GoteborgTor. | 11.79°E | 57.68°N | 0.9092 | 0.0925 | 0.9138 | 0.0908 |
| Stenungsund | 11.83°E | 58.09°N | 0.9079 | 0.1143 | 0.9115 | 0.1128 |
| Smogen | 11.22°E | 58.35°N | 0.9259 | 0.0907 | 0.9287 | 0.0895 |
| Kungsvik | 11.13°E | 58.99°N | 0.9349 | 0.1172 | 0.9365 | 0.1165 |
| OlandsNorraUdde | 17.10°E | 57.37°N | 0.9434 | 0.0633 | 0.9509 | 0.0600 |
| LandsortNorra | 17.86°E | 58.77°N | 0.9673 | 0.0698 | 0.9723 | 0.0679 |
| Skagen | 10.60°E | 57.72°N | 0.9262 | 0.0983 | 0.9271 | 0.0981 |




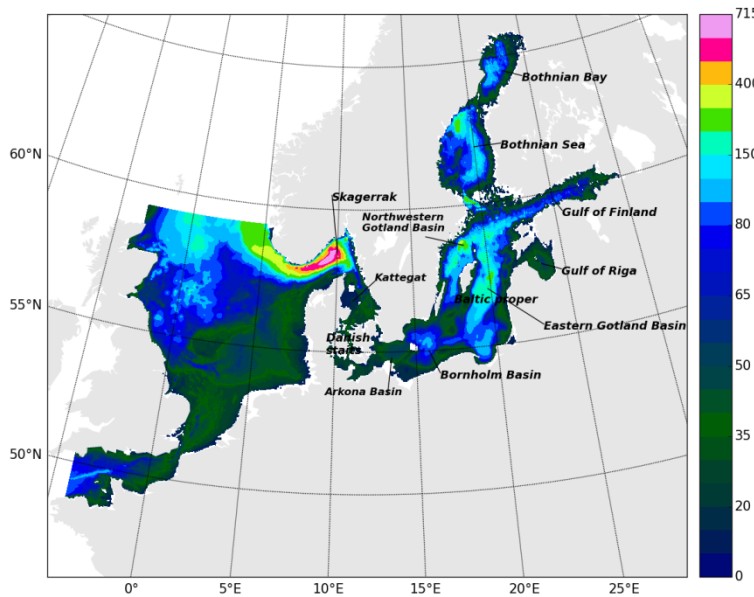

Figure 1. Geographical domain and bathymetry (in m) of the NEMO-Nordic configuration.





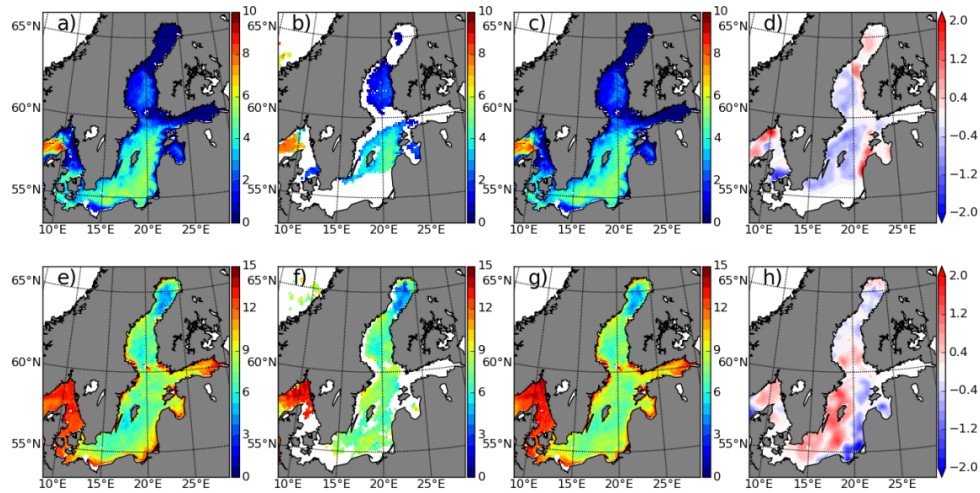

Figure 2. Map of SST from FREE (a,e), OSISAF (b, f), ASSIM (c, g) and the assimilation increments (d, h) on

11 January 2010 (first row) and 2 June 2010 (second row), respectively.



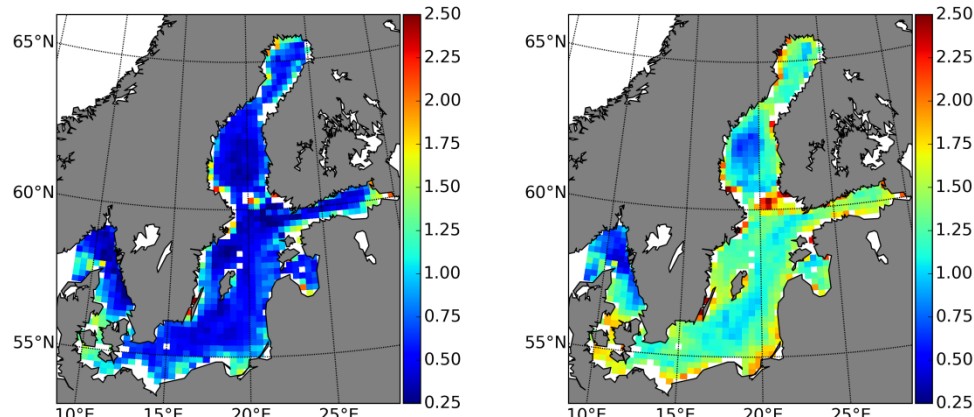

Figure 3. Map of the RMSE of SST from ASSIM (left panel) and FREE (right panel) calculated against IceMap

SST in 2010, respectively.




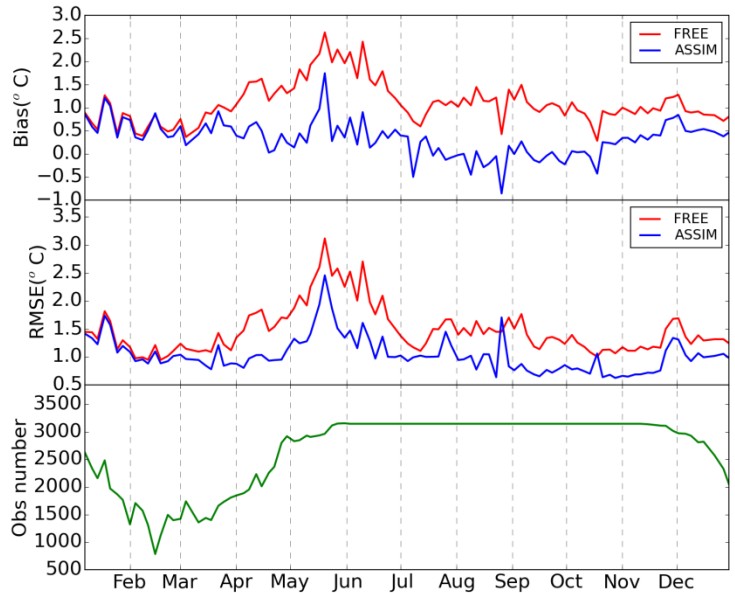

Figure 4. The evolution of basin-averaged bias and RMSE of SST from FREE and ASSIM relative to IceMap

SST and the number of IceMap observation in 2010.




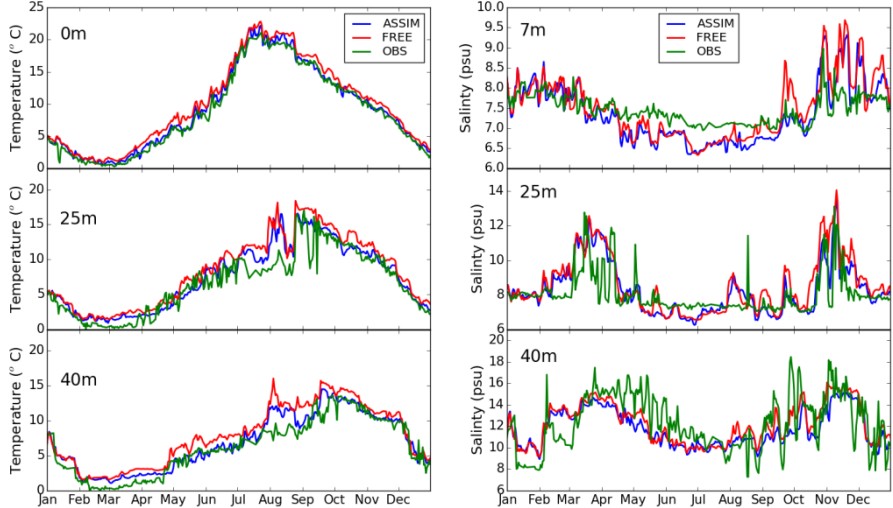

Figure 5. The time series of temperature (left panel) at a depth of 0, 25 and 40 m and salinity (right panel) at a

depth of 7, 25 and 40 m at the Arkona station (13.87°E, 54.88°N), respectively.




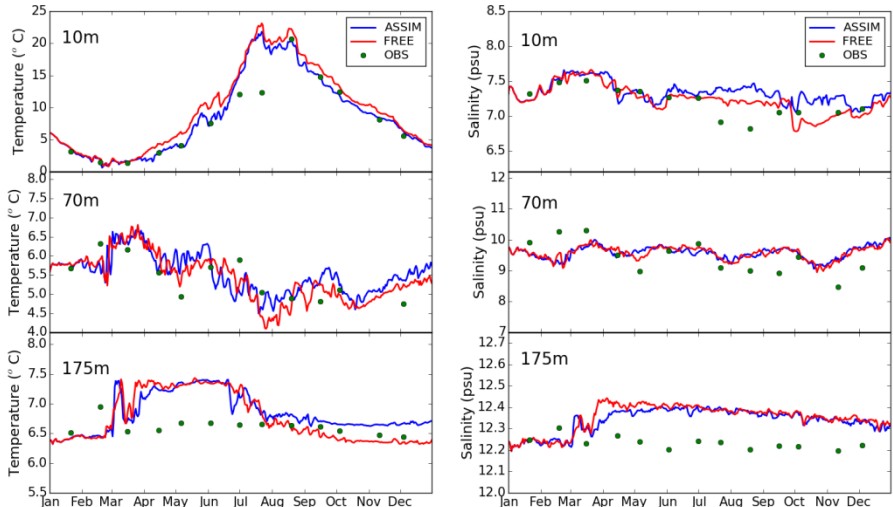

Figure 6. The time series of temperature (left panel) and salinity (right panel) at the BY15 station (20.05ºE,

57.33ºN) at a depth of 10, 70 and 175 m, respectively.





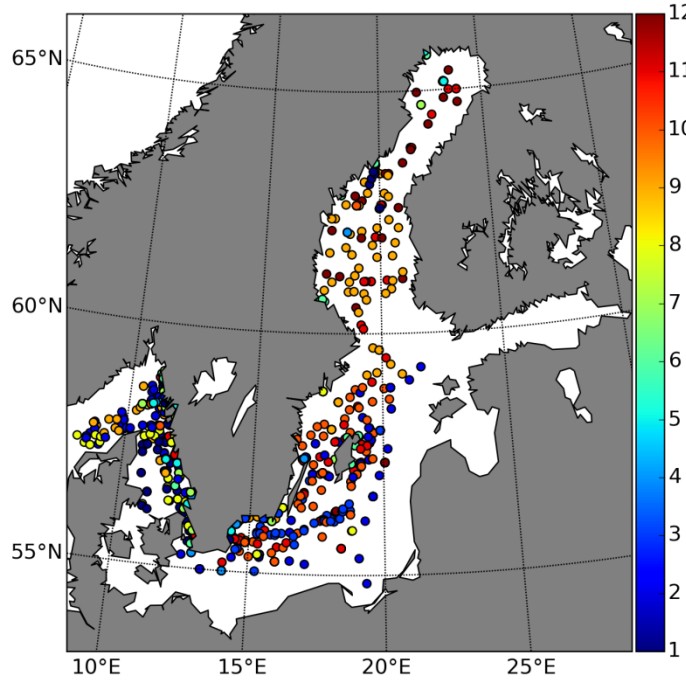

Figure 7. Map of the temperature and salinity profiles from SHARK database in 2010. The colors show the observations months.




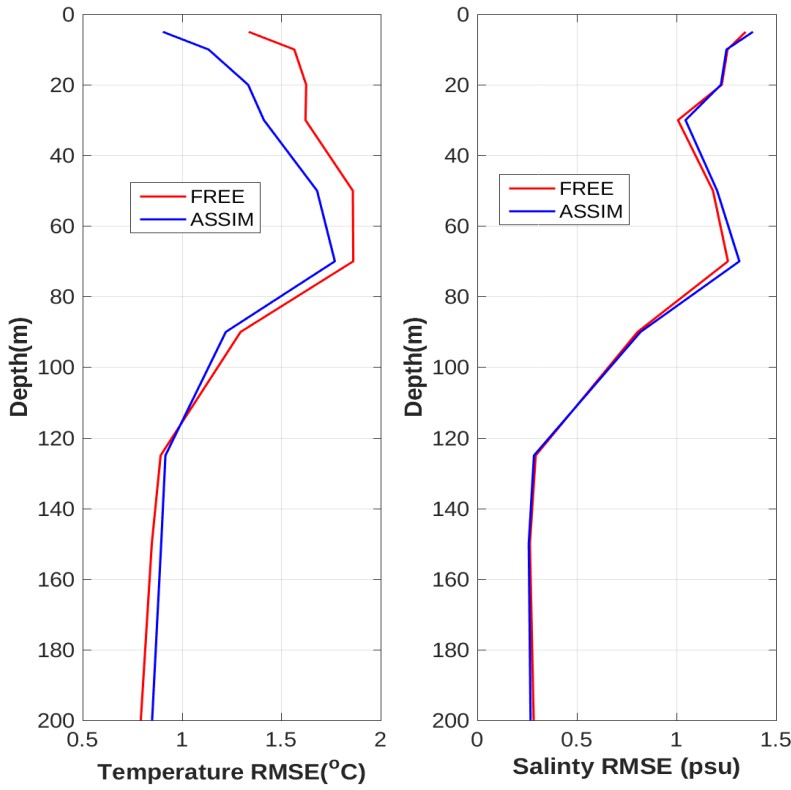

Figure 8. The overall RMSE of temperature and salinity from FREE and ASSIM relative to observations as a function of water depth.





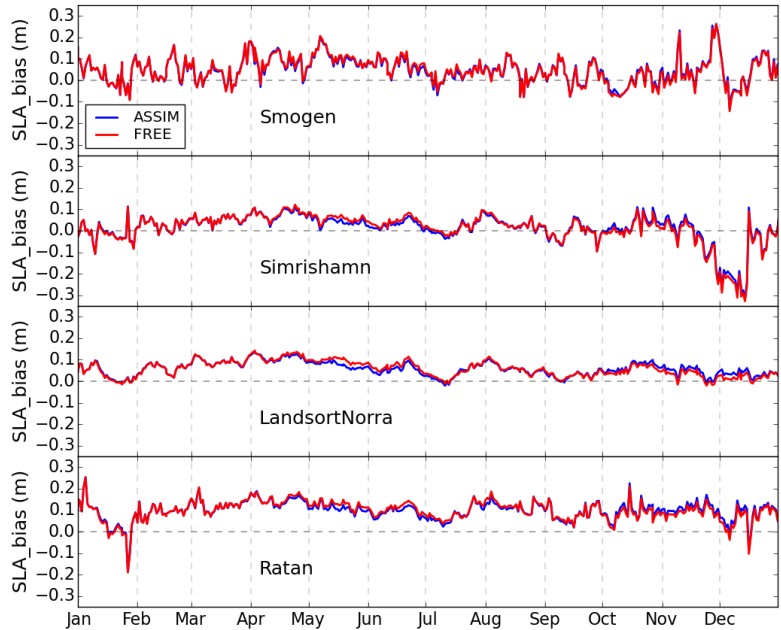

Figure 9. The biases of SLA from FREE and ASSIM relative to observations as a function of time. The position

is in the Table 1.



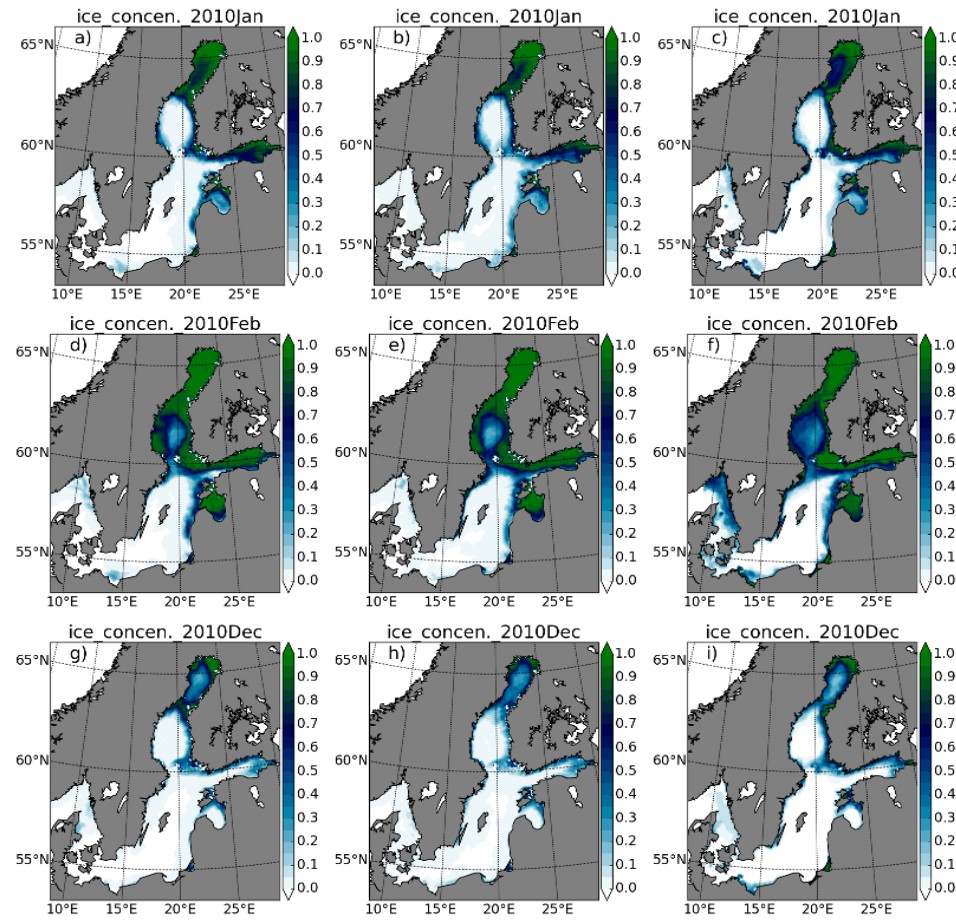

Figure 10. The monthly mean sea ice concentrations for January, February and December in FREE (left panel),

ASSIM (middle panle) and IceMap (right panel), respectively.