# Peer review of "Assimilating High-resolution Sea Surface Temperature Data"

_Ocean Science, 2018_

## Referee Comment (RC1) · Anonymous Referee #1 · 27 Mar 2018

The marine status of the Baltic Sea is highly variable and influenced by the forcing from atmosphere and freshwater influx due to shallow topography and semi-enclosed restriction. As a stable observation source, the high-resolution SST from satellite is rather important to improve the ocean operational forecast to serve the Baltic industry needs. The article of "Assimilating High-resolution Sea Surface Temperature Data Improves the Ocean Forecast in the Baltic Sea" use a localized Singular Evolutive Interpolation Kalman (SEIK) filter to assimilate the OSISAF SST during one year of 2010. Compared with dependent and independent observations, the evaluation of the model runs with and without assimilating the SST shows the SST modeling has been improved clearly. This study is suitable for publication in OS, but there are still some

obvious defects like the experimental illustration is not clear, lack of conclusions or analysis methods to inspire the readers.

The main comments are listed as follow:

1) In this study, only to assimilate the OSISAF SST in the Baltic sea. In fact, there are more SST candidates with equivalent high-resolution like OSITA (CMEMS) and RTG_SST_HR (http://polar.ncep.noaa.gov/sst/rtg_high_res/). So if assimilating one or two additional SST products, the related results will be more help the reader to well understand about them. On the other word, the special features about the OSISAF SST in the Baltic Sea have not been highlighted at current, which looks not to support the study focused on it.

2) Lines 163-165, this SST product from AVHRR is available twice daily. It is not clear how to assimilate in the experiment. The assimilation time window is daily? How to calculate the innovation, is it asynchronous?

3) In the first paragraph of 3.1, the assimilated SST has been filtered by the quality. But it is not clear how to consider the sea ice. Do you use the sea ice concentration of OSISAF to mask the SST product, and how to do?

4) The observation error for the OSISAF SST is important for this study, is it a constant of 0.5 degree used? As a good consistence check, some diagnostic about the assimilation stability like Rodwell et al. (2016) is beneficial to understand the system reliability and the observation error.

Rodwell, M. J., Lang, S. T. K., Ingleby, N. B., Bormann, N., Hólm, E., Rabier, F., Richardson, D. S., and Yamaguchi, M.: Reliability in ensemble data assimilation, Q. J. Roy. Meteor. Soc., 142, 443–454, doi:10.1002/qj.2663, 2016.

5) The IceMap has been used for evaluation as one independent SST observation. It is not objective and only twice for one week. In fact, another surface water temperature data set from SMHI collected by Ferry

(http://www.smhi.se/hfa_coord/BOOS/Ferrybox/BSNI/BSNI-Wtemp.png) is more useful and independent for this study.

6) The two in situ observations at Arkona and BY15 is super case to show the impact of assimilating SST only. It is valuable to do more specific analysis by diagnosing dynamic variables. Firstly, investigating the mixed layer depth in the two runs can clearly show the mixing strength for Fig.5 and Fig.6. Secondly, the temp/salinity misfits in vertical can be shown and mutual authentication with the SHARK results.

7) Based on the current results, it indicates the salinity looks no remarkable improvement. However, the salinity peak in Sep 2010 at 7 m can be reduced by assimilation even this model run has an underestimation before. This event is a nice case to explore which factor contributes that positive correction.

8) Fig8 shows the vertical impact for temp/saln. It is better to separate into two parts internal and out of Baltic sea.

9) The impact on SLA looks very small so I suggest replacing the related figure and table by a short paragraph.

10) Fig. 10 shows an improvement by assimilating SST. But the quantitatively comparison with the OSISAF concentration in the time series is helpful to know the impact in different sea ice seasons.

Other small issues:

1) Line 137, the operator of Li in Eq. 3 has no illustration.

2) Line 159, "OSISAF product" is it means more general products or only SST?

3) Line 229, "model layer" replaced by "model level" because the model is not a layered model.

4) Line 233, the forgotten factor is constant, or how to be defined?

5) Line 257, the evolution of SST based on 48-hourly local analysis. Does it mean all the SST comparison afterward use the 48 hourly forecast from the model?

6) Fig 1, the text is hard to identify. It is better to show the rivers involved in the model. The two stations of Arkona and BY15 can be shown in Fig. 1 (or Fig. 7).

7) Fig 6, the observed temperature at 70 m looks missing at Nov 2014, especially compared with other two depths or the salinity.

8) Line 289, the obvious improvement in the Gulf of Finland. However, based on the snapshot of the observed SST distribution in Fig. 2 there are no observations.

9) Line 278, "The model SST forecasts in both winter and summer (Fig2)". It is not corrected to say SST forecast in Fig. 2 because they only show the analyzed fields and the related increments, which not supports this conclusion.

10) Line 311, "The temperatures differ by about 15-22C between summer and winter" is confused. Does it mean the seasonal variability in observation?

11) Line 314, "The reason perhaps . . ." this kind of illustrations in this study require some proofs like MLD diagnosing or others.

12) Section 5.1, which mean ssh fields are used for tide gauges and the model simulations?

---

## Referee Comment (RC2) · S. Losa (Referee) · 9 Apr 2018

This paper describes an experience in assimilating a high-resolution satellite sea surface temperature (SST) product into a NEMO-based numerical model of the Baltic Sea. There is no doubt that the Baltic Community urgently requires a good quality forecast of the Baltic Sea hydrography. And there is no doubt that any forecasting system would definitely benefit from assimilating observational information. These circumstances provide a strong motivation for the study discussed. I am, however, not sure that this data assimilation experience is sufficient enough to be documented in a peer-reviewed publication (at least with respect to improving forecast, as stated). In its present form,

[Figure]

I cannot recommend the manuscript for publication. Below I list my comments the authors might want to consider.

General comments

The Title does not correctly reflect the subject/results of the paper. My concern is the statement "Improves the Ocean Forecast": 1) because of the use of atmospheric reanalysis (not the forecast) as the forcing; and 2) since it is not clear from the text whether the authors evaluate the system state after the LSEIK analysis or in the forecast phase (just before the analysis). Same is to the statements in lines 20 – 21, 86, 278, 440. "...the Ocean Forecast in the Baltic Sea" sounds a bit odd.

There is a lack of detailed information on the data assimilation set up: whether the ensemble error statistics (or ensemble of model trajectories) dynamically evolve(s) in time or there is just one model trajectory and at the analysis step (every 48 hours) a constant (as it looks like given the expression "a stationary ensemble sample" in line 465, the suggestion on "a flow-dependent background error covariance" in line 472) covariance matrix represents the model error statistics... The SEIK and LSEIK are normally considered as ensemble-based data assimilation methods. It would be nice if the authors clearly emphasize what is different/distinct in their application and why they use (L)SEIK for the analysis while they do not use any ensemble at all. Why do the authors not use the flow-dependent background error covariance? Do the authors really "use a localized Singular Evolutive Interpolated Kalman (SEIK) filter" just only "to characterize correlation scales in the coastal regions"? Please describe the model variables used to construct the multivariate error covariance matrix and included in the state vector.

In the present form the conclusions include only general statements on the impact of SST DA, which does not, however, add anything new to what was drawn from previous studies, and there is nothing specific with respect to assimilating the OSISAF SST. More emphasize could be made on benefits due to the resolution of the OSISAF SST

product, then a comparison against similar experiments but assimilating satellite SST data with coarser resolution are required.

Specific comments

Lines 12-13: overall the sentence sounds misleading; moreover, for the localised SEIK you can use the LSEIK abbreviation. Missing reference to Nerger et al. (2006).

Line 20 (also line 453): I am just wondering whether 0.4% difference is a statistically significant in this particular application.

Line 119: please provide a reference to the used "runoff database".

Lines 205, 206: the discussed is the representation error (Janjić et al. 2017, https://doi.org/10.1002/qj.3130).

Part 4, Lines 224-225, 228: Please explicitly determine the state vector – which particular model variables it includes.

Lines 227-228: editing is required for the sentence "There does not exist uniform nature of error covariance for the variables of the model state vector and for the coastal zones . . ."

Line 233: "a forgetting factor" or "the so-called forgetting factor"

Liner 236: missing references to Janjić′ et al. 2011

Lines 247-248: The sentence "The correlation length scale . . .." is a copy-paste from Losa et al. 2012; please rephrase and provide the references, including the references to the original studies by reporting on the estimates of the Rossby radius of deformation (Alenius et al., 2003; Fennel et al., 1991).

Lines 271-276: the discussion on the bias seasonality: while, in general, the statement (l. 271) is true and was also discussed in Losa et al. 2014, it is difficult (if ever necessary) to conclude anything in this respect given just 2 snapshots for the increments

(Figure 2).

Line 457: "significantly improved" – this is not obvious.

Misprints/Typos

Line 14: should it be "improvements of" instead of "improvements on"?

Lines 33-35: please provide references;

Line 38: "a numerical model" instead of "a numeric model";

Line 46: "joint effort" instead of "joints effort";

Line 49: "used for the operational" instead of "used to the operational"?

Line 85: "sea level anomaly" instead of "sea level Anomaly";

Line 271: "model forecast possibility" – please remove "possibility";

Line 308, 333, 335: "Arkona" instead of "Arokna";

Line 329: "The possible reason" not "The possibility reason";

Line 470: "strongly", however the sentence in the lines 470-471 sounds misleading.

References

Janjić′, T., Nerger, L., Albertella, A., Schröter, J., Skachko, S., 2011. On domain localization in ensemble based Kalman filter algorithms. Monthly Weather Review 136 (7), 2046–2060.

Nerger, L., Danilov, S., Hiller, W., Schröter, J., 2006. Using sea level data to constrain a finite-element primitive-equation model with a local SEIK filter. Ocean Dynamics 56, 634–649.

---

## Author Comment (AC1) · 2 May 2018

The marine status of the Baltic Sea is highly variable and influenced by the forcing from atmosphere and freshwater influx due to shallow topography and semi-enclosed restriction. As a stable observation source, the high-resolution SST from satellite is rather important to improve the ocean operational forecast to serve the Baltic industry needs. The article of "Assimilating High-resolution Sea Surface Temperature Data Improves the Ocean Forecast in the Baltic Sea" use a localized Singular Evolutive Interpolation Kalman (SEIK) filter to assimilate the OSISAF SST during one year of 2010. Compared with dependent and independent observations, the evaluation of the model runs with and without assimilating the SST shows the SST modeling has been improved clearly. This study is suitable for publication in OS, but there are still some obvious defects like the experimental illustration is not clear, lack of conclusions or analysis methods to inspire the readers.

We appreciate the referee for the good comments, which definitely contributes to the improvement of this study. Our responses are in blue.

The main comments are listed as follow:

1) In this study, only to assimilate the OSISAF SST in the Baltic sea. In fact, there are more SST candidates with equivalent high-resolution like OSITA (CMEMS) and RTG_SST_HR (http://polar.ncep.noaa.gov/sst/rtg_high_res/). So if assimilating one or two additional SST products, the related results will be more help the reader to well understand about them. On the other word, the special features about the OSISAF SST in the Baltic Sea have not been highlighted at current, which looks not to support the study focused on it.

We thank the reviewer for this comment. We used the OSISAF in this study for a couple of reasons. First, it is level 2 product and is retrieved directly from the satellite, which means there

is no hind-analysis information included; Second, the OSISAF has a high resolution in the Baltic Sea, which makes it more suitable for our operational forecast system.

In the revised manuscript, we added a few sentences to clarify:

"For operational forecast, the SST from OSISAF is the most important dataset in the Baltic Sea because it differs from hindcast analyzed product like OSTIA (Operational SST and Sea Ice Analysis) data. As a level 2 product, the OSISAF SST has both good temporal and spatial coverage in the Baltic Sea. As there is no hindcast information included in the OSISAF SST, we are able to assess direct impacts of assimilating SST observations"

2) Lines 163-165, this SST product from AVHRR is available twice daily. It is not clear how to assimilate in the experiment. The **a**ssimilation time window is daily? How to calculate the innovation, is it asynchronous?

In section 3.1, we mention "only the subskin SST at night, which is comparable to in situ (buoy) measurement, is used …".

In the revision, we clarified how to calculate the innovation: "Further, we define a two-day assimilation window in the assimilation experiment. As a result, the observations in the two days before the assimilation time were used to calculate the innovation with observation operator. When we calculated the innovation we also changed the observation error according to the observation time by

$$\varepsilon = 0.4 \times \exp(-0.15\Delta t) \qquad (9),$$

here $\Delta t$ is the absolute time difference between observation time and DA time."

3) In the first paragraph of 3.1, the assimilated SST has been filtered by the quality. But it is not clear how to consider the sea ice. Do you use the sea ice concentration of OSISAF to mask the SST product, and how to do?

We didn't use the sea ice concentration of OSISAF to mask the SST product. By the quality filter, we checked observation position, innovation relative to model result and the quality flag provided by OSISAF. If the model is covered by sea ice, the SST observation will be excluded.

4) The observation error for the OSISAF SST is important for this study, is it a constant of 0.5 degree used? As a good consistence check, some diagnostic about the assimilation stability like Rodwell et al. (2016) is beneficial to understand the system reliability and the observation error.

Rodwell, M. J., Lang, S. T. K., Ingleby, N. B., Bormann, N., Hólm, E., Rabier, F., Richardson, D. S., and Yamaguchi, M.: Reliability in ensemble data assimilation, Q. J. Roy. Meteor. Soc., 142, 443–454, doi:10.1002/qj.2663, 2016.

We agree that consistency check and assimilation stability are important for operational forecast systems with DA. We used a constant observation error similar to Rodwell et al. (2016) in this study, but our DA design is different from that paper. The major difference between these two studies is that we estimate the background error covariance from stationary ensembles and avoid the perturbation of observation error. Therefore, the diagnostic of the assimilation stability can be directly obtained from the forecast error, like the RMSE, in Fig.4, which shows comparable bias and RMSE in the assimilation and free forecast.

In the revision, we cited the Rodwell et al. (2016) and discussed the assimilation stability in section 6.

The corresponding text are added:

"Further, the reliability of the DA system is worth being assessed. In Rodwell et al. (2006), a perfect reliable system error variance for ensemble assimilation was calculated by the sum of the variance of the sample ensemble, the square of innovation (misfit between observation and model), the variance of observation at assimilation time. In this study, we used a constant observation error similar to Rodwell et al. (2016) because our DA design is different from that paper. The major difference between these two studies is that we estimate the background error covariance from stationary ensembles and avoid the perturbation of observation error. Therefore, the variance of the sampled ensemble and observation is univariate and the diagnostic of the assimilation stability can be directly obtained from the forecast error like the RMSE in Fig.4."

5) The IceMap has been used for evaluation as one independent SST observation. It is not objective and only twice for one week. In fact, another surface water temperature data set from SMHI collected by Ferry (http://www.smhi.se/hfa_coord/BOOS/Ferrybox/BSNI/BSNI-Wtemp.png) is more useful and independent for this study.

We agree that Ferrybox data is a very good source for model evaluation. In this study, we aim to evaluate the overall impact of OASIF SST product on the model forecast in the Baltic Sea. In this sense, the IceMap data is more preferable due to its spatial coverage and quality while the Ferrybox data has limited spatial coverage. At the same time, we have also used the independent in situ SHARK observations to verify the experiment results. The Ferrybox data may corroborate our conclusions but we think it is not a critical factor for our evaluation and conclusions.

6) The two in situ observations at Arkona and BY15 is super case to show the impact of assimilating SST only. It is valuable to do more specific analysis by diagnosing dynamic variables. Firstly, **i**nvestigating the mixed layer depth in the two runs can clearly show the mixing strength for Fig.5 and Fig.6. Secondly, the temp/salinity misfits in vertical can be shown and mutual authentication with the SHARK results.

We thank the reviewer for this important comment. To address the reviewer's comment, we compared the mixed layer depth in the two runs (Fig. 7) in the revised manuscript. We also used the SHARK data to examine the misfits of temperature and salinity at both inside and outside of the Baltic Sea(Fig.9).

In section 5.2, we added

"The mixed layer depth (MLD) was calculated at the Arkona and BY15 station and compared with the SHARK observation in Fig. 7. We used the temperature criterion to define the MLD, i.e., the depth at which the temperature deviated from the surface value by 0.5 $^{o}$C (Fu et al., 2012). Figure 7 shows that the MLD at Arkona had larger variability relative to the MLD at BY15. The reason contributed to this feature is that the deeper water at Arkona is easy affected by wind forcing because of the shallow bathymetry and well mixing, whereas the temperature

variation in upper water at BY15 difficulty influences the deeper water because of the strong stratification. Both runs had reproduced the MLD variability feature similar as the observations. For example, the minimum MLD appeared in summer, which was about several meters. The assimilation of satellite SST caused strong changes in the MLD at both stations, especially in winter. One explanation was that the Baltic Sea was largely affected by wind forcing and the winter wind was much stronger than the summer wind. Further, strong heating in summer promoted stratification in summer and shoaled the MLD."

7) Based on the current results, it indicates the salinity looks no remarkable improvement. However, the salinity peak in Sep 2010 at 7 m can be reduced by assimilation even this model run has an underestimation before. This event is a nice case to explore which factor contributes that positive correction.

We appreciate the reviewer's comment, but it is hard to attribute the improvement in September 2010 to a specific factor. There are a couple of reasons for this: firstly, at the depth of 7 m, the model salinity was strongly affected by the simulation of advection, mixing and E-P flux. Bias in any of these factors could contribute to the large bias especially after mid-September. In other words, any improvement of these factors also helped to correct the salinity bias. Secondly, the salinity at 7 m is generally decreased irrespective of the model bias, suggesting that the method is stable. Therefore, it is very likely that the improvement is a cumulative effect of our data assimilation, including the effect of the changes of circulation and mixing (shown in the mixed layer depth in Fig. 7).

8) Fig8 shows the vertical impact for temp/saln. It is better to separate into two parts internal and out of Baltic sea.

We separate the Bias and RMSE calculation in the two regions now. The figure caption of Fig. 8 was changes as Fig. 9.

[Figure]

Figure 9. The overall RMSE and bias of temperature (up panel) and salinity (down panel) from FREE and ASSIM relative to observations as a function of water depth inside (b,d) and outside (a,c) of the Baltic Sea.

The corresponding text are changed:

"Figure 9 shows the change of overall bias and RMSE of T/S with depth against the SHARK dataset. In the Baltic Sea, DA had large impact on the temperature forecast in the water above 100 m. The RMSE showed that the forecast of temperature was obviously improved from surface to thermocline in the ASSIM and the improvements generally decreased with depth. Above 100 m, the overall RMSE of temperature in ASSIM was decreased by 21.38% (from 1.59 to 1.25 °C). It was also found the temperature error had similar variability as the warm biases in two runs. In the transition zone, the RMSE in the ASSIM was reduced by 5.59% and -

20.31% above and below 100 m relative to the FREE, respectively. Below 90 m, the temperature was also over-adjusted, which changed the warm bias to cold bias. It is worth noting that the number of the deeper water observation in the transition zone is substantially less than that in the Baltic Sea. For the salinity, both RMSE and bias of the ASSIM showed very minor changes relative to the FREE inside the Baltic Sea. For the water above 100 m, the total RMSE of salinity was increased by 3.48% (from 1.15 psu in the FREE to 1.19 psu in the ASSIM) in the transition zone and 1.04% (from 0.96 psu in the FREE to 0.97 psu in the ASSIM) in the Baltic Sea."

9) The impact on SLA looks very small so I suggest replacing the related figure and table by a short paragraph.

We thank your good comment. We removed the Table 1 and added a Figure to show the variation by DA.

"We calculated the RMSE and correlation coefficients for both the FREE and ASSIM against the observations from tide gauges (Fig. 10). The overall RMSE was reduced by 1.8% and the correlation coefficients were slightly increased. Among the stations, RMSE at the Oskarshamn was decreased by 5.6%, which is larger than that at other station. The minimum RMSE change of SLA was seen at the Klagshamn. For the correlation coefficient, improvement on the SLA by the DA is very small. Simrishamn station showed the biggest change of correlation coefficient, which is 1.1%. The RMSE and correlation comparison demonstrated that the SST DA has generally positive effects on the forecast of the SLA."

[Figure]

Figure 10. The improvement (%) of correlation coefficient and RMSE for the SLA at 10 tide gauges stations. The positions are shown in Fig. 8b.

Further, we also replaced the old figure 9 by Figure 11 to show the bias variation after data assimilation.

[Figure]

Figure 11. The difference of SLA biases between ASSIM and FREE against observations as a function of time at four observing stations.

10) Fig. 10 shows an improvement by assimilating SST. But the quantitatively comparison with the OSISAF concentration in the time series is helpful to know the impact in different sea ice seasons.

We added a new figure showing the comparison of monthly mean sea ice concentration in March and April and we also added the time series of the sea ice extent (SIE).

In the manuscript, we revised the text as:

"In March, compared to observation, the FREE produced low SIC in the western coast of the Bothnian Sea, Gulf of Finland, Gulf of Riga and the connect zone between the Bothnian Sea and Gulf of Finland. However, the model SIC in the FREE was higher than IceMap in the interior the Bothnian Bay. For instance, the SIC from FREE in the western Bothnian Sea was 40% higher than observation. In the south coast of the Arkona basin and Baltic proper, the FREE failed to reproduce the sea ice as in observation. After the DA, the high SIC was decreased in western Bothnian Sea and closer to that in IceMap in Bothnian Sea. In the Gulf of Finland and Gulf of Riga, the SIC error was increased in the ASSIM. In April, the large SIC error in the FREE was shown in the Bothnian Sea, the Bothnian Bay, Gulf of Rig and Gulf of Finland, where no clear improvements were seen in the ASSIM."

"The daily SIE from the FREE and ASSIM was compared with observations in Fig.13. The observed SIE was generally increased from January to February and reached the maximum in mid-February. During the period of March-May, SIE was decreased as temperature was increasing. SIEs in both the FREE and ASSIM experiments were generally underestimated by comparison with the observation in 2010, especially in the period from Mid-March to early April. The SIE bias in both runs was increased from January to early April. In early April, the maximum negative bias of SIE was found to be 105000 $km^2$ for the ASSIM and 10000 $km^2$ for the FREE. The impact of SST assimilation on the SIE was positive during the phase of sea ice formation. For example, the SIE bias was reduced 25000 $km^2$ at the end of February and in the Mid-December. However, during the phase of sea ice melting (March to April), SIE error was increased in the ASSIM even with the error of SST decreased. For example, the SIE bias in the

ASSIM was increased by 42000 km² relative to FREE in the early March. These increased SIE

error in March mainly happened in the Gulf of Riga and Gulf of Finland (Fig.11)."

[Figure]

Figure 13. The daily sea ice extent from FREE, ASSIM and IceMap and the sea ice extent bias

(modelled minus observed field), respectively.

Other small issues:

1) Line 137, the operator of Li in Eq. 3 has no illustration.

We added the line for the operator **L** illustration.

2) Line 159, "OSISAF product" is it means more general products or only SST?

To clarify, we delete " products are using in priority the European Meteorological satellites

METEOSAT and MetOp and also several American satellites operated by NOAA, DMSP and

NASA. Its"

3) Line 229, "model layer" replaced by "model level" because the model is not a layered model.

It was corrected.

4) Line 233, the forgotten factor is constant, or how to be defined?

We add a sentence " To define the forgetting factor, a one-month simulation experiment with

varying the factor ρ was done in January 2010. At last, a factor ρ = 0.3 resulted in the best

assimilation performance." At the end of Section 4.

5) Line 257, the evolution of SST based on 48-hourly local analysis. Does it mean all the SST comparison afterward use the 48 hourly forecast from the model?

Yes, we use the 48-hour forecast SST in the all comparison with observation.

6) Fig 1, the text is hard to identify. It is better to show the rivers involved in the model. The two stations of Arkona and BY15 can be shown in Fig. 1 (or Fig. 7).

We add the Neva River and the position of Arkona and BY15 in Fig.1

7) Fig 6, the observed temperature at 70 m looks missing at Nov 2014, especially compared with other two depths or the salinity.

This temperature at 70 at BY15 station hasn't observation value at Nov 2010 in SHARK database.

8) Line 289, the obvious improvement in the Gulf of Finland. However, based on the snapshot of the observed SST distribution in Fig. 2 there are no observations.

The OSISAF observation at a specific basin may be missing like the Figure 2. Our Figure 3 is based on annual averaged IceMap SST comparison in 2010.

9) Line 278, "The model SST forecasts in both winter and summer (Fig2)". It is not corrected to say SST forecast in Fig. 2 because they only show the analyzed fields and the related increments, which not supports this conclusion.

Thank you. We changed it to "the SST DA has improved the simulated SST in both cases (Fig.2)"

10) Line 311, "The temperatures differ by about 15-22C between summer and winter" is confused. Does it mean the seasonal variability in observation?

We intended to show the seasonal variability. Since we only done one-year simulation, we delete the sentence "The temperatures differ by about 15–22 $^o$C between summer and winter." to avoid confusion.

11) Line 314, "The reason perhaps ..." this kind of illustrations in this study require some proofs like MLD diagnosing or others.

We add the mixed layer depth analysis, which will support our conclusion.

12) Section 5.1, which mean ssh fields are used for tide gauges and the model simulations?

Since the mean SSH fields may be different from each other in the model and observation we do the comparison of SLA in this study. We calculated the mean SSH by directly averaging the tide gauges or model fields.

---

## Author Comment (AC3) · 2 May 2018

S. Losa (Referee)

svetlana.losa@awi.de

This paper describes an experience in assimilating a high-resolution satellite sea surface temperature (SST) product into a NEMO-based numerical model of the Baltic Sea. There is no doubt that the Baltic Community urgently requires a good quality forecast of the Baltic Sea hydrography. And there is no doubt that any forecasting system would definitely benefit from assimilating observational information. These circumstances provide a strong motivation for the study discussed. I am, however, not sure that this data assimilation experience is sufficient enough to be documented in a peer-reviewed publication (at least with respect to improving forecast, as stated). In its present form, I cannot recommend the manuscript for publication. Below I list my comments the authors might want to consider.

We thank the referee for reviewing our manuscript, acknowledging the importance of our topic, and for her suggestions on how to improve the manuscript.

General comments

The Title does not correctly reflect the subject/results of the paper. My concern is the statement "Improves the Ocean Forecast": 1) because of the use of atmospheric reanalysis (not the forecast) as the forcing; and 2) since it is not clear from the text whether the authors evaluate the system state after the LSEIK analysis or in the forecast phase (just before the

analysis). Same is to the statements in lines 20 – 21, 86, 278, 440. "the Ocean Forecast in the Baltic Sea" sounds a bit odd.

1) We changed the title a bit to "Assimilating High-resolution Sea Surface Temperature Data Improves the Ocean Forecast Potential in the Baltic Sea". We think the new title more correctly reflect the subject as our main focus is to demonstrate the potential impact of SST assimilation on the model predictions of the NEMO-Nordic, which aims for operational forecast. We believe that the reanalysis forcing will not impair this objective of our paper. In this study, we intended to showcase how useful the DA method and SST dataset are within the framework of NEMO-Nordic. One advantage of the reanalysis forcing is to reduce the uncertainty/bias of results arising from NWP forcing. We chose the atmospheric reanalysis to force the model, which, we thought, may reduce the intervention of the atmospheric errors regarding the analysis of the experiment results.

2) In the revision, we clarified this issue by pointing out that the system evaluation was done in the forecast phase after the LSEIK filter analysis (Fig.2). In this study, we firstly verified the system with two cases in two different seasons. Then we assess the impact of the SST data assimilation on Baltic forecast based on 48-hours forecast.

In the first paragraph of section 5, we added

"*We considered the evolution of SST based on 48-hourly local analysis from 1 January 2010 to 31 December 2010. The 48-hourly forecast of SST from two runs was assessed with observations from different dataset (see Fig. 2 for details).*"

There is a lack of detailed information on the data assimilation set up: whether the ensemble error statistics (or ensemble of model trajectories) dynamically evolve(s) in time or there is just one model trajectory and at the analysis step (every 48 hours) a constant (as it looks like given the expression "a stationary ensemble sample" in line 465, the suggestion on "a flow-dependent background error covariance" in line 472) covariance matrix represents the model

error statistics: The SEIK and LSEIK are normally considered as ensemble-based data assimilation methods. It would be nice if the authors clearly emphasize what is different/distinct in their application and why they use (L)SEIK for the analysis while they do not use any ensemble at all. Why do the authors not use the flow-dependent background error covariance? Do the authors really "use a localized Singular Evolutive Interpolated Kalman (SEIK) filter" just only "to characterize correlation scales in the coastal regions"? Please describe the model variables used to construct the multivariate error covariance matrix and included in the state vector.

According the reviewer's comment, we added more details to clarify these points. we used a stationary ensemble to statistically estimate the background error covariance. We did not use time-varying ensemble based on a couple of considerations: 1) firstly, the stationary ensemble is computationally efficient as we don't need to integrated many model states like the EnKF. Secondly, the time-invariant ensemble was shown to be able to mimic the signature of circulation in the background error covariance (Fu et al., 2011; Liu et al., 2013). Time-varying vs time-invariant ensemble is an interesting topic with respect to approximating the background error covariance (Korre et al. 2004). However, the major objective of this study is to validate the assimilation of high resolution SST data. Given the number of ensemble samples used in this study and our previous study, we are confident that the stationary ensemble can produce robust analysis (Liu et al. 2013).

We added the text in the revision:" We used a time-invariant sample ensemble to approximate the background error covariance during the experimental period (Korres et al, 2004, Liu et al. 2013, 2017). This stationary ensemble affords a good approximation of the ocean's background error covariance. Meanwhile, it is computationally efficient for our objective."

We described the LSEIK in more details in the revised manuscript.

Similar to other ensemble data assimilation EnOI or EnKF, the SEIK filter method includes both the global and local analysis based on different consideration. We used local analysis version of SEIK (LSEIK) with domain localization in this study (Neger et al. 2007). We used a localization scale of 70km for the Baltic and North Sea. Now we moved the following text from Section 4 to Section 2.2:

"*Localization was used to remove the unrealistic long-range correlation with a quasi-Gaussian function and a uniform horizontal correlation scale (Liu et al. 2013). It was performed by neglecting observations that were beyond correlation distance from an analyzed grid point. In other words, only data located in the "neighborhood" of an analyzed grid point contribute to the analysis at this point*."

We stated that the state vector includes the sea level, temperature and salinity. The same model variables (sea level, temperature and salinity) were also used in the multivariable EOF analysis.

To further clarify the DA setup, we also added the text about the observation error and forget factor: "To define the forgetting factor, a one-month simulation experiment with varying the factor $\rho$ was done in January 2010. At last, a factor $\rho = 0.3$ resulted in the best assimilation performance. Further, we define a two-day assimilation window in assimilation experiment. As a result, the observations in the two days before the assimilation time were used to calculate the innovation with observation operator. When we calculated the innovation we also changed the observation error according to the observation time by

$$\varepsilon = 0.4 \times \exp(-0.15\Delta t) \qquad (9),$$

here $\Delta t$ is the absolute time difference between observation time and DA time. "

In the present form the conclusions include only general statements on the impact of SST DA, which does not, however, add anything new to what was drawn from previous studies, and there is nothing specific with respect to assimilating the OSISAF SST. More emphasize could be

made on benefits due to the resolution of the OSISAF SST product, then a comparison against similar experiments but assimilating satellite SST data with coarser resolution are required.

The major objective of this study is to demonstrate the potential impact of assimilating OSISAF SST product on the forecast of the Baltic Sea. We showed in detail the potential of SST data assimilation for the forecast of temperature, salinity, sea level, mixed layer depth and sea ice. This study provides a clear and informative image to the Baltic Sea community for improving the forecast of different fields in the future. It is also the first time that OSISAF SST was assimilated into NEMO-Nordic model, which will replace the old operational forecasting system and serve the operational purposes at SMHI.

We summarized some new results in this study:

1.  We demonstrated the potential of SST assimilation for the Baltic Sea forecast with the OSISAF Level 2 product, which is not contaminated by hind-cast information.

2.  We provided overall validations of the potential impact of SST assimilation for the forecast in the whole Baltic Sea (both shallow basins and much deeper regions such as the Gotland Basin).

3.  We found that the assimilation of SST could generally improve the forecasts of sea level from late spring to summer.

4.  We showed an in-depth evaluation of the impact of SST assimilation on sea ice forecast by comparing the model with the observations of sea ice concentration (SIC) and sea ice extent (SIE). We found that the impact of SST assimilation on sea ice forecast is time-dependent, more important during the phase of sea ice formation than sea ice melting (March-April).

The assimilation of coarser resolution of the OSISAF product into the same model is interesting, but we would respectfully think it is beyond the scope of this study. Actually,

previous studies showed that proper 'observation-thinning' schemes were very helpful to assimilate high-density remote sensing data. For instance, Li et al (2009) assimilated 0.3ºx0.3º satellite SST observation in the Chinese shelf-coastal seas. With an ensemble-based observation-thinning scheme, the assimilation of coarser resolution SST (0.5ºx0.5º) can yield an Analysis Error variance (AEV) of 0.1ºC. In the Baltic Sea, we expect that the impact of coarsening SST data on the forecast is weakened to some degree, depending on the actual thinning scheme.

Li XC, Zhu J, Xiao YG, Wang RW (2010) A Model-Based Observation-Thinning Scheme for the Assimilation of High-Resolution SST in the Shelf and Coastal Seas around China Journal of Atmospheric and Oceanic Technology 27:1044-1058 doi:10.1175/2010jtecho709.1

Specific comments

Lines 12-13: overall the sentence sounds misleading; moreover, for the localised SEIK you can use the LSEIK abbreviation. Missing reference to Nerger et al. (2006).

Thank you. We now used the LSEIK for the data assimilation method.The Nerger et al. (2006) was added as a reference in Section 2.2.

Line 20 (also line 453): I am just wondering whether 0.4% difference is a statistically significant in this particular application.

The model SLA was highly correlate with observation. The improvement of SLA varies considerably with stations.  The 0.4% difference is the overall impact of the SST assimilation on the SLA.  Since it is difficult to test the significance of the overall impact, we removed this sentence.

Line 119: please provide a reference to the used "runoff database".

We added the reference for the river runoff data:

Donnelly, C., Andersson, J. C., and Arheimer, B.: Using flow signatures and catchment similarities to evaluate the E-HYPE multi-basin model across Europe, Hydrological Sciences Journal, 61, 255–273, 2016.

Lines 205, 206: the discussed is the representation error (Janji´c et al. 2017, https://doi.org/10.1002/qj.3130).

There is different definition of the components of observation error in different consideration and theory. For clarify, we removed the discussion "The observation error mainly comes from the observation instrument itself, the observation representativeness, the temporary reading error and imperfect retrieval algorithm."

Part 4, Lines 224-225, 228: Please explicitly determine the state vector – which particular model variables it includes.

In the reversion, we added one sentence to clarify:

 The sate vector includes sea level, temperature and salinity.

Lines 227-228: editing is required for the sentence "There does not exist uniform nature of error covariance for the variables of the model state vector and for the coastal zones "

Thank you. We rephrased the sentence as : "In the North Sea and Baltic Sea, error covariances of different variables are not uniform and strongly dependent on whether the variable resides in the open sea or coastal zone."

Line 233: "a forgetting factor" or "the so-called forgetting factor"

Thank you. we use "the forgetting factor" as the same using in Nerger et al. (2006).

Liner 236: missing references to Janji´c0 et al. 2011

We added this reference Janjić et al. (2011)

Lines 247-248: The sentence "The correlation length scale : : :." is a copy-paste from

Losa et al. 2012; please rephrase and provide the references, including the references

to the original studies by reporting on the estimates of the Rossby radius of deformation

(Alenius et al., 2003; Fennel et al., 1991).

we rephrased this sentence and added the Losa et al., (2012) as a reference.

"The correlation length scale is to some extent dependent on the Rossby radius of deformation

(Losa et al., 2012), which varies from ~ 200 km in the barotropic mode to ~ 10 km or even less

in the baroclinic mode (Fennel et al., 1991; Alenius et al, 2003)."

Lines 271-276: the discussion on the bias seasonality: while, in general, the statement

(l. 271) is true and was also discussed in Losa et al. 2014, it is difficult (if ever necessary)

to conclude anything in this respect given just 2 snapshots for the increments (Figure 2).

Thanks for good comment. We removed the seasonally bias discussion related to Figure 2.

Line 457: "significantly improved" – this is not obvious.

we removed "significantly".

Line 14: should it be "improvements of" instead of "improvements on"?

we revised it to "improvements".

Lines 33-35: please provide references;

we added a reference Omstedt et al. 2014:

Omstedt, A., Elken, J., Lehmann,A., Leppäranta, M., Meier, H.E.M., Myrberg, K., and

Rutgersson, A.: Progress in physical oceanography of the Baltic Sea during the 2003–2014

period. Progress in Oceanography, 128, 139-171, 2014.

Line 38: "a numerical model" instead of "a numeric model";

It was fixed.

Line 46: "joint effort" instead of "joints effort";

It was fixed.

Line 49: "used for the operational" instead of "used to the operational"?

It was fixed.

Line 85: "sea level anomaly" instead of "sea level Anomaly";

It was fixed.

Line 271: "model forecast possibility" – please remove "possibility";

In revision, we don't want to discuss the season variation of model SST. Therefore, we deleted the sentence "The SST bias of model forecast **possibility** has seasonal variability because of the errors in the forcing and/or heat flux parameterization used in the ocean model (Fu et al. 2012)."

Line 308, 333, 335: "Arkona" instead of "Arokna";

It was fixed.

Line 329: "The possible reason" not "The possibility reason";

It was fixed.

Line 470: "strongly", however the sentence in the lines 470-471 sounds misleading.

We removed "around the observation position."

References

Janji´c0, T., Nerger, L., Albertella, A., Schröter, J., Skachko, S., 2011. On domain localization

in ensemble based Kalman filter algorithms. Monthly Weather Review 136 (7),

2046–2060.

Nerger, L., Danilov, S., Hiller, W., Schröter, J., 2006. Using sea level data to constrain

a finite-element primitive-equation model with a local SEIK filter. Ocean Dynamics 56,634–649.

---

## Author Comment (AC4) · 2 May 2018

The comment was uploaded in the form of a supplement:
https://www.ocean-sci-discuss.net/os-2018-8/os-2018-8-AC4-supplement.pdf

---

## Author Response (AR2)

We'd like to thank the Dr. S. Losa for her useful comments to further improve our manuscript. Our responses are in blue.

I propose excluding all the equations (1-7) from Part 2.2 and equation 8 from Part 4 from the text since they were already published by Pham et al. and the presented manuscript is not on the method.

As an alternative, the authors might want to move them in an Appendix.

1) According the reviewer's comment, we moved the text according to the equations 1-7 from Part 2 to the appendix.  The corresponding text has been changed to:

*"The LSEIK filter produces the correction for the model state by weighting the difference between the observations and the model state estimation. The weight coefficients are constructed by the model error covariance matrix and observation error covariance matrix. Similar as other ensemble data assimilation methods, the LSEIK filter uses the spread of sample ensemble to estimate the uncertainties of the model state. Further, a forgetting factor ρ is introduced to parameterize the imperfect model by amplifying the already existing modes of the background error (Pham et al. 1998; Pham, 2001). Furthermore, the LSEIK filter is based on an explicit low-rank approximation of the model error covariance matrix. A second-order exact sampling method is used to initialize the LSEIK filter (Pham, 2001)."*

2) We also removed the equation 8. The texts associated  with equation 9 is changed to

"… according to the observation time by $\varepsilon = 0.4 \times \exp(-0.15\Delta t)$, here $\Delta t$ is the absolute time difference between observation time and DA time."

-Lines 13 - 14: the sentence "We use ... (LSEIK) filter to characterize correlation scales in the coastal regions" could be rewritten in a way that... the authors use a Kalman-type filtering to assimilate the data..., and they use a low rank approximation of the stationary background error covariance metrics used at the analysis steps.

In revision, we changed the text to "*We use a Kalman-type filtering to assimilate the observations in the coastal regions. Further, a low rank approximation of the stationary background error covariance metrics is used at the analysis steps.*"

- Lines 230 - 232: accounting for the representation error is indeed important. Please see Janjić et al 2017 (the study also addresses the terminology issue).

Janjić, T. , Bormann, N. , Bocquet, M. , Carton, J. A., Cohn, S. E., Dance, S. L., Losa, S. N., Nichols, N. K., Potthast, R. , Waller, J. A. and Weston, P. (2017), On the representation error in data assimilation. Q.J.R. Meteorol. Soc.. . doi:10.1002/qj.3130

We added some texts to account for the measurement error and observation error in the Section 3.2:

"*The error for an observation used in data assimilation mainly includes the representation error and the measurement error. The measurement error arises primarily from the measuring instruments, the temporary reading error and imperfect retrieval algorithm. According to Janjić et al. (2017), the representation error in data assimilation comprises the error due to unsolved scales or processes, the pre-processing error and the observation-operator error.*"

- Lines 266 - 267: the correct references with respect to forgetting factor are Pham et al. (1998a,b) and Pham 2001.

Thank you. We updated the references to Pham et al. 1998b and Pham 2001 as suggested.

Pham, D. T., Verron, J. and L. Gourdeau, 1998a: Filtres de Kalman singuliers évolutif pour l'assimilation de données en océnographie. Compt. Rend. Acad. Sci. Terre Planètres,326, 255–260.

[revised manuscript text omitted]